# $p$-Poisson surface reconstruction in curl-free flow from point clouds

**Yesom Park[1]\*, Taekyung Lee[2]\*, Jooyoung Hahn[3], Myungjoo Kang[1]**
[1] Department of Mathematical Sciences, Seoul National University
[2] Interdisciplinary Program in Artificial Intelligence, Seoul National University
[3] Department of Mathematics and Descriptive Geometry,
Slovak University of Technology in Bratislava
{yeisom, dlxorud1231, mkang}@snu.ac.kr
jooyoung.hahn@stuba.sk

## Abstract

The aim of this paper is the reconstruction of a smooth surface from an unorganized point cloud sampled by a closed surface, with the preservation of geometric shapes, without any further information other than the point cloud. Implicit neural representations (INRs) have recently emerged as a promising approach to surface reconstruction. However, the reconstruction quality of existing methods relies on ground truth implicit function values or surface normal vectors. In this paper, we show that proper supervision of partial differential equations and fundamental properties of differential vector fields are sufficient to robustly reconstruct high-quality surfaces. We cast the $p$-Poisson equation to learn a signed distance function (SDF) and the reconstructed surface is implicitly represented by the zero-level set of the SDF. For efficient training, we develop a variable splitting structure by introducing a gradient of the SDF as an auxiliary variable and impose the $p$-Poisson equation directly on the auxiliary variable as a hard constraint. Based on the curl-free property of the gradient field, we impose a curl-free constraint on the auxiliary variable, which leads to a more faithful reconstruction. Experiments on standard benchmark datasets show that the proposed INR provides a superior and robust reconstruction. The code is available at https://github.com/Yebbi/PINC.

## 1 Introduction

Surface reconstruction from an unorganized point cloud has been extensively studied for more than two decades [10, 29, 40, 9, 62] due to its many downstream applications in computer vision and computer graphics[8, 16, 62, 58]. Classical point cloud or mesh-based representations are efficient but they do not guarantee a watertight surface and are usually limited to fixed geometries. Implicit function-based representations of the surface [28, 64, 43, 14] as a level set $\mathcal{S} = \left\{ \mathbf{x} \in \mathbb{R}^3 \mid u(\mathbf{x}) = c \right\}$ of a continuous implicit function $u : \mathbb{R}^3 \to \mathbb{R}$, such as signed distance functions (SDFs) or occupancy functions, have received considerable attention for providing watertight results and great flexibility

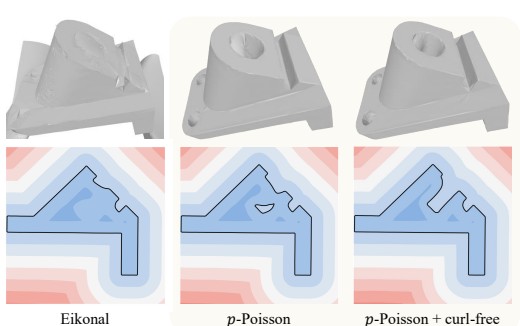

Eikonal      $p$-Poisson      $p$-Poisson + curl-free

Figure 1: Comparison of reconstruction using an eikonal equation (9), the $p$-Poisson equation (8), and the proposed $p$-Poisson equation with the curl-free condition (11).

---

\*Equal contribution authors. Correspondence to: <mkang@snu.ac.kr>.

37th Conference on Neural Information Processing Systems (NeurIPS 2023).

in representing different topologies. In recent years, with the rise of deep learning, a stream of work called *implicit neural representations* (INRs) [2, 44, 16, 61, 19, 55, 53, 50] has revisited them by parameterizing the implicit function $u$ with neural networks. INRs have shown promising results by offering efficient training and expressive surface reconstruction.

Early INRs [44, 42, 16] treat the points-to-surface problem as a supervised regression problem with ground-truth distance values, which are difficult to use in many situations. To overcome this limitation, some research efforts have used partial differential equations (PDEs), typically the eikonal equation, as a means to relax the 3D supervision [23, 37, 48]. While these efforts have been successful in reconstructing various geometries, they encounter an issue of non-unique solutions in the eikonal equation and rely heavily on the oriented normal vector at each point. They often fail to capture fine details or reconstruct plausible surfaces without normal vectors. A raw point cloud usually lacks normal vectors or numerically estimated normal vectors [1, 18] contain approximation errors. Moreover, the prior works are vulnerable to noisy observations and outliers.

The goal of this work is to propose an implicit representation of surfaces that not only provides smooth reconstruction but also recovers high-frequency features only from a raw point cloud. To this end, we provide a novel approach that expresses an approximated SDF as the unique solution to the $p$-Poisson equation. In contrast to previous studies that only describe the SDF as a network, we define the gradient of the SDF as an auxiliary variable, motivated by variable splitting methods [47, 60, 22, 12] in the optimization literature. We then parameterize the auxiliary output to automatically satisfy the $p$-Poisson equation by reformulating the equation in a divergence-free form. The divergence-free splitting representation contributes to efficient training by avoiding deeply nested gradient chains and allows the use of sufficiently large $p$, which permits an accurate approximation of the SDF. In addition, we impose a curl-free constraint [25] because the auxiliary variable should be learned as a conservative vector field which has vanishing curl. The curl-free constraint serves to achieve a faithful reconstruction. We carefully evaluate the proposed model on widely used benchmarks and robustness to noise. The results demonstrate the superiority of our model without a priori knowledge of the surface normal at the data points.

## 2 Background and related works

**Implicit neural representations** In recent years, implicit neural representations (INRs) [41, 16, 3, 55, 54], which define a surface as zero level-sets of neural networks, have been extensively studied. Early work requires the ground-truth signed implicit function [44, 16, 41], which is difficult to obtain in real-world scenarios. Considerable research [3, 4] is devoted to removing 3D supervision and relaxing it with a ground truth normal vector at each point. In particular, several efforts use PDEs to remove supervision and learn implicit functions only from raw point clouds. Recently, IGR [23] revisits a conventional numerical approach [14] that accesses the SDF by incorporating the eikonal equation into a variational problem by using modern computational tools of deep learning. Without the normal vector, however, IGR misses fine details. To alleviate this problem, FFN [56] and SIREN [55] put the high frequencies directly into the network. Other approaches exploit additional loss terms to regulate the divergence [6] or the Hessian [63]. The vanishing viscosity method, which perturbs the eikonal equation with a small diffusion term, is also considered [37, 49] to mitigate the drawback that the eikonal loss has unreliable minima. The classical Poisson reconstruction [31], which recovers the implicit function by integration over the normal vector field, has also been revisited to accelerate the model inference time [48], but supervision of the normal vector field is required. Neural-Pull [39] constructs a new loss function by borrowing the geometrical property that the SDF and its gradient define the shortest path to the surface.

**$p$-Poisson equation** The SDF is described by a solution of various PDEs. The existing work [23, 55, 6] uses the eikonal equation, whose viscosity solution describes the SDF. However, the use of the residual of the eikonal equation as a loss function raises concerns about the convergence to the SDF due to non-unique solutions of the eikonal equation. Recent works [55, 6] utilize the notion of vanishing viscosity to circumvent the issue of non-unique solutions. In this paper, we use the $p$-Poisson equation to approximate the SDF, which is a nonlinear generalization of the Poisson equation ($p = 2$):

$$\begin{cases} -\triangle_p u = -\nabla \cdot \left( \|\nabla u\|^{p-2} \nabla u \right) = 1 \text{ in } \Omega \\ u = 0 \text{ on } \Gamma, \end{cases} \tag{1}$$

where $p \geq 2$, the computation domain $\Omega \subset \mathbb{R}^3$ is bounded, and $\Gamma$ is embedded in $\Omega$.

The main advantage of using the $p$-Poisson equation is that the solution to (1) is unique in Sobolev space $W^{1,p}(\Omega)$ [36]. The unique solution with $p \geq 2$ brings a viscosity solution of the eikonal equation in the limit $p \to \infty$, which is the SDF, and it eventually prevents finding non-viscosity solutions of the eikonal equation; see a further discussion with an example in Appendix C.1. Moreover, in contrast to the eikonal equation, it is possible to describe a solution of (1) as a variational problem and compute an accurate approximation [5, 20]:

$$\min_u \int_\Omega \frac{\|\nabla u\|^p}{p} d\mathbf{x} - \int_\Omega u d\mathbf{x}. \tag{2}$$

As $p \to \infty$, it has been shown [11, 30] that the solution $u$ of (1) converges to the SDF whose zero level set is $\Gamma$. As a result, increasing $p$ gives a better approximation of the SDF, which is definitely helpful for surface reconstruction. However, it is still difficult to use a fairly large $p$ in numerical computations and in this paper we will explain one of the possible solutions to the mentioned problem.

## 3 Method

In this section, we propose a $p$-**P**oisson equation based **I**mplicit **N**eural representation with **C**url-free constraint (**PINC**). From an unorganized point cloud $\mathcal{X} = \{\mathbf{x}_i : i = 1, 2, \ldots, N\}$ sampled by a closed surface $\Gamma$, a SDF $u : \mathbb{R}^3 \to \mathbb{R}$ whose zero level set is the surface $\Gamma = \{\mathbf{x} \in \mathbb{R}^3 \mid u(\mathbf{x}) = 0\}$ is reconstructed by the proposed INR. There are two key elements in the proposed method: First, using a variable-splitting representation [45] of the network, an auxiliary output is used to learn the gradient of the SDF that satisfies the $p$-Poisson equation (1). Second, a curl-free constraint is enforced on an auxiliary variable to ensure that the differentiable vector identity is satisfied.

### 3.1 $p$-**Poisson equation**

A loss function in the physics-informed framework [51] of the existing INRs for the $p$-Poisson equation (1) can be directly written:

$$\min_u \int_\Gamma |u| \, d\mathbf{x} + \lambda_0 \int_\Omega \left| \nabla \cdot \left( \|\nabla u\|^{p-2} \nabla u \right) + 1 \right| d\mathbf{x}, \tag{3}$$

where $\lambda_0 > 0$ is a regularization constant. To reduce the learning complexity of the second integrand, we propose an augmented network structure that separately parameterizes the gradient of the SDF as an auxiliary variable that satisfies the $p$-Poisson equation (1).

**Variable-splitting strategy**   Unlike existing studies [23, 37, 6] that use neural networks with only one output $u$ for the SDF, we introduce a separate auxiliary network output $G$ for the gradient of the SDF; see that the same principle is used in [45]. In the optimization literature, it is called the variable splitting method [47, 60, 22, 12] and it has the advantage of decomposing a complex minimization into a sequence of relatively simple sub-problems. With the auxiliary variable $G = \nabla u$ and the penalty method [13], the variational problem (3) is converted into an unconstrained problem:

$$\min_{u, G} \int_\Gamma |u| \, d\mathbf{x} + \lambda_0 \int_\Omega \left| \nabla \cdot \left( \|G\|^{p-2} G \right) + 1 \right| d\mathbf{x} + \lambda_1 \int_\Omega \|\nabla u - G\|^2 \, d\mathbf{x}, \tag{4}$$

where $\lambda_1 > 0$ is a penalty parameter representing the relative importance of the loss terms.

$p$-**Poisson as a hard constraint**   Looking more closely at the minimization (4), if $G$ is already a gradient to satisfy (1), then the second term in (4) is no longer needed and it brings the simplicity of one less parameter. Now, for a function $F : \Omega \to \mathbb{R}^3$ such that $\nabla \cdot F = 1$, for example $F(\mathbf{x}) = \frac{1}{3}\mathbf{x}$, the $p$-Poisson equation (1) is reformulated by the divergence-free form:

$$\nabla \cdot \left( \|\nabla u\|^{p-2} \nabla u + F \right) = 0. \tag{5}$$

Then, there exists a vector potential $\Psi : \mathbb{R}^3 \to \mathbb{R}^3$ satisfying

$$\|G\|^{p-2} G + F = \nabla \times \Psi, \tag{6}$$

where $G = \nabla u$. Note that a similar idea is used in the neural conservation law [52] to construct a divergence-free vector field built on the Helmholtz decomposition [33, 57]. From the condition (6), we have $\|G\|^{p-1} = \|\nabla \times \Psi - F\|$ and $G$ is parallel to $\nabla \times \Psi - F$, then the auxiliary output $G$ is explicitly written:

$$G = \frac{\nabla \times \Psi - F}{\|\nabla \times \Psi - F\|^{\frac{p-2}{p-1}}}. \tag{7}$$

This confirms that the minimization problem (4) does not require finding $G$ directly, but rather that it can be obtained from the vector potential $\Psi$. Therefore, the second loss term in (4) can be eliminated by approximating the potential function $\Psi$ by a neural network and defining the auxiliary output $G$ as a hard constraint (7). To sum up, we use a loss function of the form

$$\mathcal{L}_{p\text{-Poisson}} = \int_\Gamma |u| \, d\mathbf{x} + \lambda_1 \int_\Omega \|\nabla u - G\|^2 \, d\mathbf{x}, \tag{8}$$

where $G$ is obtained by (7), the first term is responsible for imposing the boundary condition of (1), and the second term enforces the constraint $G = \nabla u$ between primary and auxiliary outputs. It is worth mentioning that $G$ in (7) is designed to exactly satisfy the $p$-Poisson equation (1).

An advantage of the proposed loss function (8) and the hard constraint (7) is that (1) can be solved for sufficiently large $p$, which is critical to make a better approximation of the SDF. It is not straightforward in (3) or (4) because the numeric value of $(p-2)$-power with a large $p$ easily exceeds the limit of floating precision. On the other hand, in (7) we use $(p-2)/(p-1)$-power, which allows stable computation even when $p$ becomes arbitrarily large. The surface reconstruction with varying $p$ in Figure 7 shows that using a large enough $p$ is crucial to get a good reconstruction. As the $p$ increases, the reconstruction gets closer and closer to the point cloud. Furthermore, it is worth noting that the proposed representation expresses the second-order PDE (1) with first-order derivatives only. By reducing the order of the derivatives, the computational graph is simplified than (3) or (4).

Note that one can think of an approach to directly solve the eikonal equation $\|\nabla u\| = 1$ with an auxiliary variable $H = \nabla u$ as an output of the neural network:

$$\min_{u, \|H\|=1} \int_\Gamma |u| \, d\mathbf{x} + \eta \int_\Omega \|\nabla u - H\|^2 \, d\mathbf{x}, \tag{9}$$

where $\eta > 0$. The above loss function may produce a non-unique solution of the eikonal equation, which causes numerical instability and undesirable estimation of the surface reconstruction; see Figure 1. To alleviate such an issue, the vanishing viscosity method is used in [37, 49] to approximate the SDF by $u_\sigma$ as $\sigma \to 0$, a solution of $-\sigma \triangle u_\sigma + \text{sign}(u_\sigma)(|\nabla u_\sigma| - 1) = 0$. However, the results are dependent on the hyper-parameter $\sigma > 0$ related to the resolution of the discretized computational domain and the order of the numerical scheme [17, 24].

### 3.2 Curl-free constraint

In the penalty method, we have to compute more strictly to ensure that $G = \nabla u$ by using progressively larger values of $\lambda_1$ in (8), but in practice we cannot make the value of $\lambda_1$ infinitely large. Now, we can think of yet another condition for enforcing the constraint $G = \nabla u$ from a differential vector identity which says a conservative vector field is curl-free:

$$\nabla \times G = 0 \iff G = \nabla u \tag{10}$$

for some scalar potential function $u$. While it may seem straightforward, adding a penalty term $\int_\Omega \|\nabla \times G\|^2 \, d\mathbf{x}$ at the top of (8) is fraught with problems. Since $G$ is calculated by using a curl operation (7), the mentioned penalty term makes a long and complex computational graph. In addition, it has been reported that such loss functions, which include high-order derivatives computed by automatic differentiation, induce a loss landscape that is difficult to optimize [34, 59]. In order to relax the mentioned issue, we augment another auxiliary variable $\tilde{G}$, where $G = \tilde{G}$ and $\nabla \times \tilde{G} = 0$ are constrained.

By incorporating the new auxiliary variable $\tilde{G}$ and its curl-free constraint, we have the following loss function:

$$\mathcal{L}_{\text{PINC}} = \mathcal{L}_{p\text{-Poisson}} + \lambda_2 \int_\Omega \left\|G - \tilde{G}\right\|^2 \, d\mathbf{x} + \lambda_3 \int_\Omega \left\|\nabla \times \tilde{G}\right\|^2 \, d\mathbf{x}. \tag{11}$$

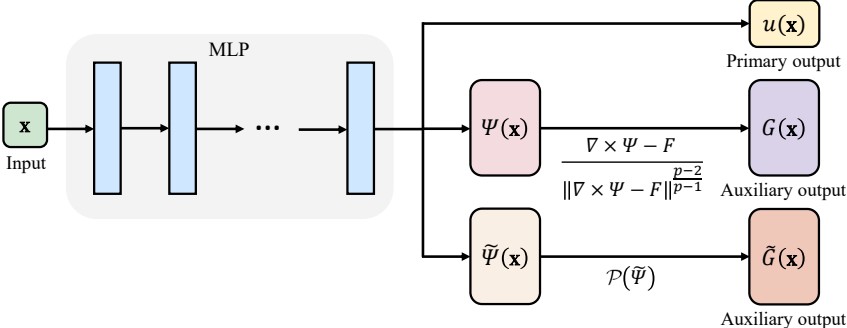

Figure 2: The visualization of the augmented network structure with two auxiliary variables.

Note that the optimal $\tilde{G}$ should have a unit norm according to the eikonal equation. To facilitate training, we relax this nonconvex equality condition into a convex constraint $\| \tilde{G} \| \leq 1$. To this end, we parameterize the second network auxiliary output $\tilde{\Psi}$ and define $\tilde{G}$ by

$$\tilde{G} = \mathcal{P}\left(\tilde{\Psi}\right) := \frac{\tilde{\Psi}}{\max\left\{1, \| \tilde{\Psi} \|\right\}}, \tag{12}$$

where $\mathcal{P}$ is the projection operator to the three-dimensional unit ball. Appendix A provides further discussion on the importance of the curl-free term to learn a conservative vector field.

Figure 2 illustrates the proposed network architecture. The primary and the auxiliary variables are trained in a single network, instead of being trained separately in individual networks. The number of network parameters remains almost the same since only the output dimension of the last layer is increased by six, while all hidden layers are shared.

### 3.3 Proposed loss function

In the case of a real point cloud to estimate a closed surface by range scanners, it is inevitable to have occluded parts of the surface where the surface has a concave part depending on possible angles of the measurement [35]. It ends up having relatively large holes in the measured point cloud. Since there are no points in the middle of the hole, it is necessary to have a certain criterion for how to fill in the hole. In order to focus to check the quality of $\mathcal{L}_{\text{PINC}}$ (11) in this paper, we choose a simple rule to minimize the area of zero level set of $u$:

$$\mathcal{L}_{\text{total}} = \mathcal{L}_{\text{PINC}} + \lambda_4 \int_{\Omega} \delta_\epsilon\left(u\right) \|\nabla u\| \, d\mathbf{x}, \tag{13}$$

where $\lambda_4 > 0$ and $\delta_\epsilon(x) = 1 - \tanh^2\left(\frac{x}{\epsilon}\right)$ is a smeared Dirac delta function with $\epsilon > 0$. The minimization of the area is used in [21, 49] and the advanced models [15, 27, 63] on missing parts of the point cloud to provide better performance of the reconstruction.

## 4 Experimental results

In this section, we evaluate the performance of the proposed model to reconstruct 3D surfaces from point clouds. We study the following questions: **(i)** How does the proposed model perform compared to existing INRs? **(ii)** Is it stable from noise? **(iii)** What is the role of the parts that make up the model and the loss? Each is elaborated in order in the following sections.

**Implementation**  As in previous studies [44, 23, 37], we use an 8-layer network with 512 neurons and a skip connection to the middle layer, but only the output dimension of the last layer is increased by six due to the auxiliary variables. For (13), we empirically set the loss coefficients to $\lambda_1 = 0.1$, $\lambda_2 = 0.0001$. $\lambda_3 = 0.0005$, and $\lambda_4 = 0.1$ and use $p = \infty$ in (7) for numerical simplicity. We implement all numerical experiments on a single NVIDIA RTX 3090 GPU. In all experiments, we use the Adam optimizer [32] with learning rate $10^{-3}$ decayed by 0.99 every 2000 iterations.

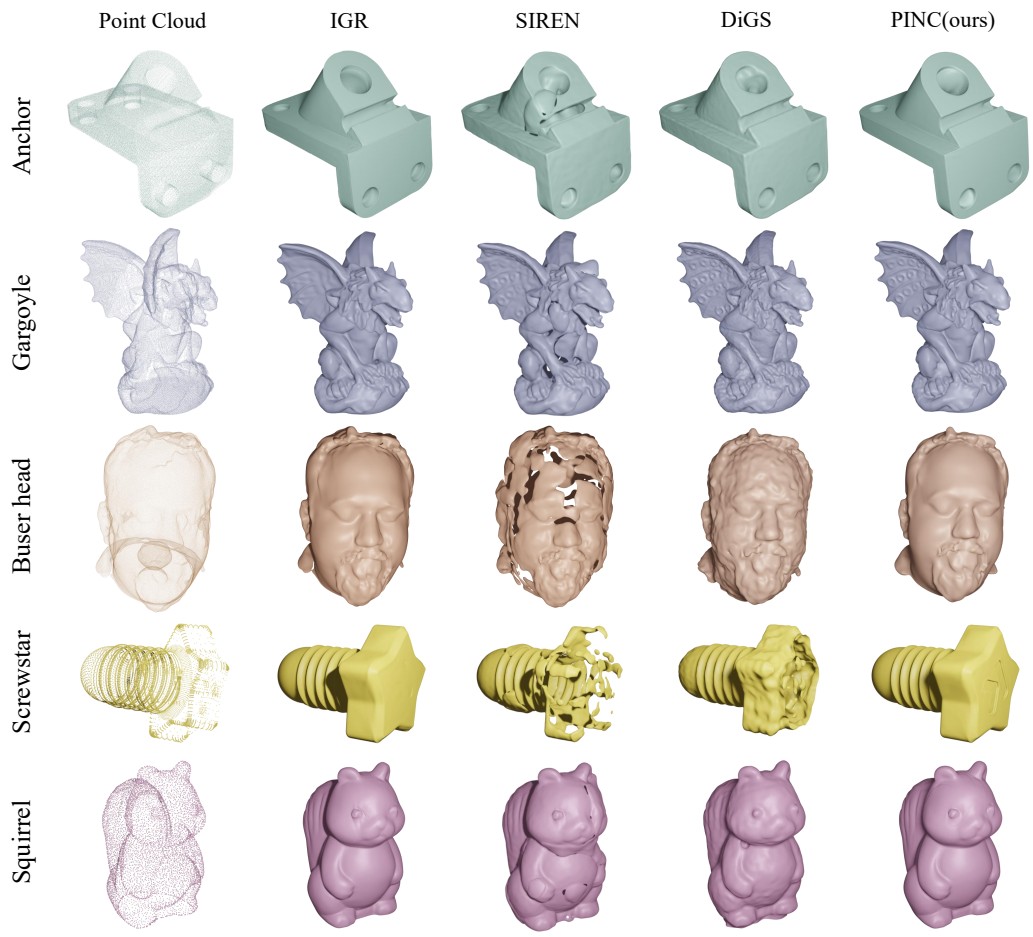

Figure 3: 3D Reconstruction results for SRB and Thingi10K datasets.

**Datasets** We leverage two widely used benchmark datasets to evaluate the proposed model for surface reconstruction: Surface Reconstruction Benchmark (SRB) [7] and Thingi10K [65]. The geometries in the mentioned datasets are challenging because of their complex topologies and incomplete observations. Following the prior works, we adopt five objects per dataset. We normalize the input data to center at zero and have a maximum norm of one.

**Baselines** We compare the proposed model with the following baselines: IGR [23], SIREN [55], SAL [3], PHASE [37], and DiGS [6]. All models are evaluated from only raw point cloud data without surface normal vectors. A comparison with models that leverage surface normals as supervision is included in Appendix C.

**Metrics** To estimate the quantitative accuracy of the reconstructed surface, we measure Chamfer $(d_C)$ and Hausdorff $(d_H)$ distances between the ground-truth point clouds and the reconstructed surfaces. Moreover, we report one-sided distances $d_{\overrightarrow{C}}$ and $d_{\overrightarrow{H}}$ between the noisy data and the reconstructed surfaces. Please see Appendix B.2 for precise definitions.

## 4.1 Surface reconstruction

We validate the performance of the proposed PINC (13) in surface reconstruction in comparison to other INR baselines. For a fair comparison, we consider the baseline models that were trained without a normal prior. Table 1 summarizes the numerical comparison on SRB in terms of metrics. We report the results of baselines from [37, 49, 6]. The results show that the reconstruction quality obtained is on par with the leading INRs, and we achieved state-or-the-art performance for Chamfer distances.

Table 1: Results on surface reconstruction of SRB.

| Model | Anchor GT $d_C$ | $d_H$ | Anchor Scans $d_{\vec{C}}$ | $d_{\vec{H}}$ | Daratech GT $d_C$ | $d_H$ | Daratech Scans $d_{\vec{C}}$ | $d_{\vec{H}}$ | DC GT $d_C$ | $d_H$ | DC Scans $d_{\vec{C}}$ | $d_{\vec{H}}$ | Gargoyle GT $d_C$ | $d_H$ | Gargoyle Scans $d_{\vec{C}}$ | $d_{\vec{H}}$ | Loard Quas GT $d_C$ | $d_H$ | Loard Quas Scans $d_{\vec{C}}$ | $d_{\vec{H}}$ |
|---|---|---|---|---|---|---|---|---|---|---|---|---|---|---|---|---|---|---|---|---|
| IGR | 0.45 | 7.45 | 0.17 | 4.55 | 4.9 | 42.15 | 0.7 | 3.68 | 0.63 | 10.35 | 0.14 | 3.44 | 0.77 | 17.46 | 0.18 | 2.04 | 0.16 | 4.22 | 0.08 | 1.14 |
| SIREN | 0.72 | 10.98 | 0.11 | 1.27 | 0.21 | 4.37 | 0.09 | 1.78 | 0.34 | 6.27 | 0.06 | **2.71** | 0.46 | 7.76 | 0.08 | **0.68** | 0.35 | 8.96 | 0.06 | **0.65** |
| SAL | 0.42 | 7.21 | 0.17 | 4.67 | 0.62 | 13.21 | 0.11 | 2.15 | 0.18 | 3.06 | 0.08 | 2.82 | 0.45 | 9.74 | 0.21 | 3.84 | 0.13 | 414 | 0.07 | 4.04 |
| PHASE | **0.29** | 7.43 | **0.09** | 1.49 | 0.35 | 7.24 | **0.08** | **1.21** | 0.19 | 4.65 | 0.05 | 2.78 | 0.17 | 4.79 | 0.07 | 1.58 | 0.11 | **0.71** | 0.05 | 0.74 |
| DiGS | **0.29** | **7.19** | 0.11 | 1.17 | **0.20** | **3.72** | 0.09 | 1.80 | 0.15 | **1.70** | 0.07 | 2.75 | 0.17 | **4.10** | 0.09 | 0.92 | 0.12 | 0.91 | 0.06 | 0.70 |
| **PINC** | **0.29** | 7.54 | **0.09** | 1.20 | 0.37 | 7.24 | 0.11 | 1.88 | **0.14** | 2.56 | **0.04** | 2.73 | **0.16** | 4.78 | **0.05** | 0.80 | **0.10** | 0.92 | **0.04** | 0.67 |

Table 2: Results on surface reconstruction of Thingi10K.

| Model | Squirrel $d_C$ | $d_H$ | Buser head $d_C$ | $d_H$ | Screwstar $d_C$ | $d_H$ | Frogrock $d_C$ | $d_H$ | Pumpkin $d_C$ | $d_H$ |
|---|---|---|---|---|---|---|---|---|---|---|
| IGR | 0.36 | 11.97 | 0.38 | 5.95 | 0.18 | 3.02 | 0.48 | 12.05 | 0.11 | **1.13** |
| SIREN | 0.47 | **5.66** | 0.43 | **4.81** | 0.27 | 4.98 | 0.78 | 14.75 | 0.46 | 5.03 |
| DiGS | 0.50 | 12.45 | 0.39 | 10.64 | 0.26 | 6.33 | 0.45 | **10.50** | 0.32 | 8.03 |
| **PINC** | **0.35** | 11.55 | **0.37** | 6.19 | **0.17** | **3.00** | **0.43** | 11.06 | **0.10** | 1.90 |

We further verify the accuracy of the reconstructed surface for the Thingi10K dataset by measuring the metrics. For Thingi10K, we reproduce the results of IGR, SIREN, and DiGS without normal vectors using the official codes. Results on Thingi10K presented in Table 2 show the proposed method achieves superior performance compared to existing approaches. PINC achieves similar or better metric values on all objects.

The qualitative results are presented in Figure 3. SIREN, which imposes high-frequency features to the model by using a sine periodic function as activation, restores a somewhat torn surface. Similarly, DiGS restores rough and rogged surfaces, for example, the human face and squirrel body are not smooth and are rendered unevenly. On the other hand, IGR provides smooth surfaces but tends to over-smooth details such as the gargoyle's wings and detail on the star-shaped bolt head of screwstar. The results confirm that the proposed PINC (13) adopts both of these advantages: PINC represents a smooth and detailed surface. More results can be found in the Appendix C.

## 4.2 Reconstruction from noisy data

In this section, we analyze whether the proposed PINC (13) produces robust results to the presence of noise in the input point data. In many situations, the samples obtained by the scanning process contain a lot of noise and inaccurate surface normals are estimated from these noisy samples. Therefore, it is an important task to perform accurate reconstruction using only noisy data without normal vectors. To investigate the robustness to noise, we perturb the data with additive Gaussian noise with mean zero and two standard deviations 0.005 and 0.01.

We quantify the ability of the proposed model to handle noise in the input points. The qualitative results are shown in Figure 4. Compared to existing methods, the results demonstrate superior resilience of the proposed model with respect to noise corruption in the input samples. We can observe that SIREN and DiGS restore broken surfaces that appear to be small grains as the noise level increases. On the other hand, the proposed model produces a relatively smooth reconstruction. Results show that PINC is less sensitive to noise than others.

## 4.3 Ablation studies

This section is devoted to ablation analyses which show that each part of the proposed loss function $\mathcal{L}_{\text{total}}$ in conjunction with the divergence-free splitting architecture plays an important role in high-quality reconstruction.

**Effect of curl-free constraint**  We first study the effect of the curl-free constraint on reconstructing high fidelity surfaces. To investigate the effectiveness of the proposed curl-free constraint, we compare the performance of PINC without the curl-free loss term, i.e., the model trained with the loss function $\mathcal{L}_{p\text{-Poisson}}$ (8). The results on the SRB dataset are reported in Table 3 and Figure 5. Figure 5

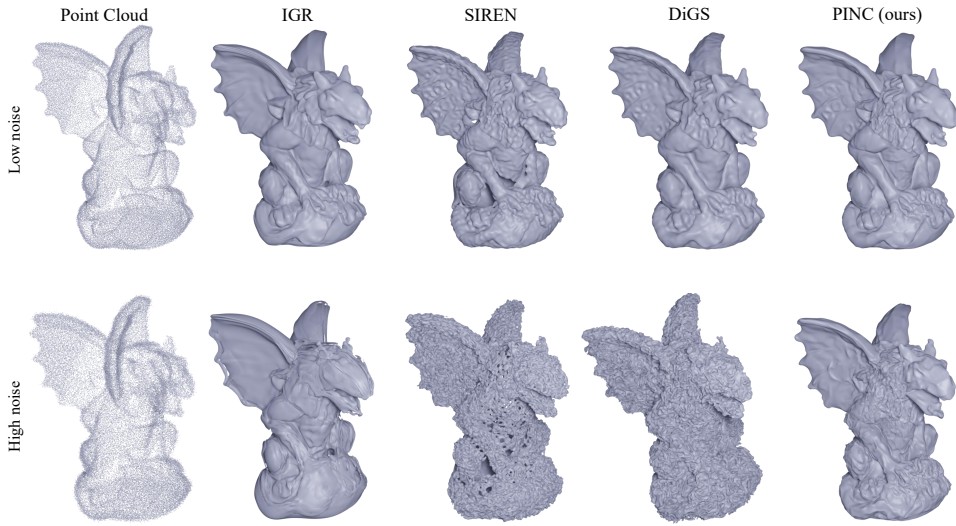

Figure 4: Reconstruction results from noisy observations. Two levels of additive Gaussian noise with standard deviations $\sigma = 0.005$ (low) and $0.01$ (high) are considered.

shows that the variable splitting method, which satisfies the $p$-Poisson equation as a hard constraint (without the curl-free condition), recovers a fairly decent surface, but it generates oversmoothed surfaces and details are lost. However, as we can see from the qualitative result reconstructed with the curl-free constraint, this constraint allows us to capture the details that PINC without the curl-free condition cannot recover. The metric values presented in Table 3 also provide clear evidence of the need for the curl-free term. To further examine the necessity of another auxiliary variable $\tilde{G}$, we conduct an additional experiment by applying the curl-free loss term directly on $G$ without the use of $\tilde{G}$. The results are presented in the second row of the Table 3. The results indicate that taking curl on $G$, which is constructed by taking curl on $\Psi$ in (7), leads to a suboptimal reconstruction. This is likely due to a challenging optimization landscape that is difficult to optimize as a result of consecutive automatic differentiation [59]. The results provide numerical evidences of the necessity of introducing $\tilde{G}$.

Table 3: Quantitative results on the ablation study of the curl-free term.

| Model | GT | | Scans | |
|---|---|---|---|---|
| | $d_C$ | $d_H$ | $d_{\vec{C}}$ | $d_{\vec{H}}$ |
| wo/ curl free | 0.20 | 4.96 | 0.12 | 2.98 |
| w/ curl free on $G$ | 4.17 | 52.26 | 0.48 | 6.03 |
| w/ curl free on $\tilde{G}$ | 0.16 | 4.78 | 0.05 | 0.80 |

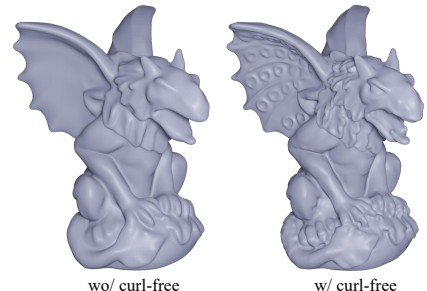

wo/ curl-free      w/ curl-free

Figure 5: Comparison of surface reconstruction without (left) and with (right) curl-free constraint.

**Effect of minimal area criterion** We study the effect of the minimal area criterion suggested in Section 3.3. In real scenarios, there are defected regions where the surface has not been measured. To fill this part of the hole, the minimum surface area is considered. Figure 6 clearly shows this effect. Some parts in the daratech of SRB have a hole in the back. Probably because of this hole, parts that are not manifolds are spread out as manifolds as shown in the left figure without considering the minimal area. However, we can see that adding a minimal area loss term alleviates this problem. We would like to note that, except for daratech, we did not encounter this problem because other data are point clouds sampled from a closed surface and also are not related to hole filling. Indeed, we

empirically observe that the results are quite similar with and without the minimal area term for all data other than daratech.

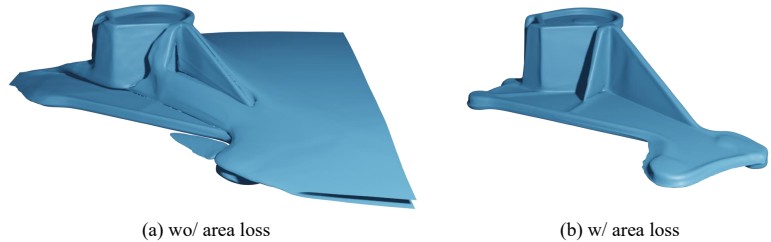

(a) wo/ area loss          (b) w/ area loss

Figure 6: Comparison of surface recovery without (a) and with (b) minimum area criterion.

**Effect of large $p$**  The $p$-Poisson equation (1) draws the SDF as $p$ becomes infinitely large. Therefore, it is natural to think that it would be good to use a large $p$. Here, we conducted experiments on the effect of $p$. We define $G$ with various $p = 2, 10$, and $100$ and learn the SDF with it. Figure 7 shows surfaces that were recovered from the Gargoyle data in the SRB with different $p$ values. When $p$ is as small as 2, it is obvious that it is difficult to reconstruct a compact surface from points. When $p$ is 10, a much better surface is constructed than that of $p = 2$, but the by-products still remain on the small holes. Furthermore, a large value of $p = 100$ provides a quite proper reconstruction. This experimental result demonstrates that a more accurate approximation can be obtained by the use of a large $p$, which is consistent with the theory. This once again highlights the advantage of the variable splitting method we have proposed, which allows an arbitrarily large $p$ to be used. This highlights the advantage of the variable splitting method (7) we have proposed in Section 3.1, which allows an arbitrarily large $p$ to be used. Note that the previous approaches have not been able to use large $p$ because the numeric value of $p$-power easily exceeds the limit of floating precision. On the other hand, the proposed method is amenable to large $p$ and hence the reconstruction becomes closer to the point cloud.

## 5 Conclusion and limitations

We presented a $p$-Poisson equation-based shape representation learning, termed PINC, that reconstructs high-fidelity surfaces using only the locations of given points. We introduced the gradient of the SDF as an auxiliary network output and incorporated the $p$-Poisson equation into the auxiliary variable as a hard constraint. The curl-free constraint was also used to provide a more accurate representation. Furthermore, the minimal surface area regularization was considered to provide a compact surface and overcome the ill-posedness of the surface reconstruction problem caused by unobserved points. The proposed PINC successively achieved a faithful surface with intricate details and was robust to noisy observations.

The minimization of the surface area is used to reconstruct missing parts of points under the assumption that a point cloud is measured by a closed surface. Regarding the hole-filling strategy, it still needs further discussion and investigation of various constraints such as mean curvature or total variation of the gradient. At present, the proposed PDE-based framework is limited to closed surfaces and is inadequate to reconstruct open surfaces. We leave the development to open surface reconstruction as future work. Establishing a neural network initialization that favors the auxiliary gradient of the SDF would be an interesting venue. Furthermore, the computational cost of convergence would differ when using and not using auxiliary variables. Analyzing the convergence speed or computational cost of utilizing auxiliary variables versus not utilizing them is a worthwhile direction for future research.

## 6 Societal Impacts

The proposed PINC allows high-quality representation of 3D shapes only from raw unoriented 3D point cloud. It has many potential downstream applications, including product design, security, medical imaging, robotics, and the film industry. We are aware that accurate 3D surface reconstruction

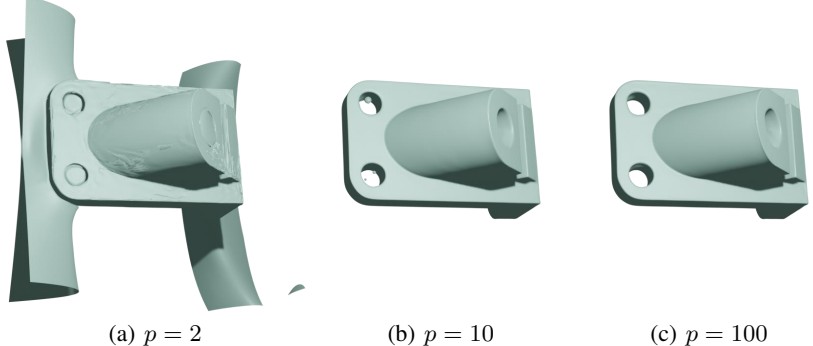

|              |              |              |
|:------------:|:------------:|:------------:|
| (a) $p = 2$  | (b) $p = 10$ | (c) $p = 100$ |

Figure 7: Surface reconstruction of anchor data with various $p$. The results show the importance of using a sufficiently large $p$ for an accurate approximation.

can be used in malicious environments such as unauthorized reproduction of machines without consent and digital impersonation. However, it is not a work to develop a technique to go to abuse, and we hope and encourage users of the proposed model to concenter on the positive impact of this work.

## 7 Acknowledgements

This work was supported by the NRF grant [2012R1A2C3010887] and the MSIT/IITP ([1711117093], [2021-0-00077], [No. 2021-0-01343, Artificial Intelligence Graduate School Program(SNU)]). Also, this project has received funding from the European Union's Horizon 2020 research and innovation programme under the Marie Skłodowska-Curie grant agreement No. 945478.

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

# A   More discussion on curl-free term

This section is devoted to both theoretically and empirically validate the necessity of the curl-free loss term. One might think that the curl-free term is unnecessary, since a curl-free $G$ can be obtained by reducing the $L^2$ penalty term for the variable splitting constraint $\nabla u = G$. However, this penalty term is not sufficient to train $G$ as a conservative vector field. In subsequent sections, we prove this theoretically and verify experimentally that this is indeed the case in practice.

## A.1   Theoretical justification

In the following theorem, we opine that minimizing the $L^2$ energy of $\|\nabla u - G\|$ in (8) without the curl-free term is not sufficient to obtain a conservative vector field $G$.

**Theorem A.1.** *There is a sequence $\{u_n, G_n\}_{n \in \mathbb{N}}$ such that $\int_\Omega \|\nabla u_n - G_n\|^2 \, d\mathbf{x} \to 0$ as $n \to \infty$, but $G_n$ does not converge to a curl-free vector field.*

*Proof.* For every $\{u_n\}_{n \in \mathbb{N}}$ defined on $\Omega = [0, 1]^3$, set

$$G_n(x, y, z) = \nabla u_n(x, y, z) + \left(0, \frac{1}{n} \sin(2\pi n x), 0\right) \in \mathbb{R}^3.$$

Then,

$$\int_\Omega \|\nabla u_n - G_n\|^2 \, d\mathbf{x} = \int_\Omega \left\|\left(0, \frac{1}{n} \sin(2\pi n x), 0\right)\right\|^2 d\mathbf{x} \tag{14}$$

$$= \frac{1}{n^2} \int_\Omega \sin^2(2\pi n x) \, d\mathbf{x} \tag{15}$$

$$\to 0, \tag{16}$$

as $n \to \infty$. However,

$$\nabla \times G_n(x, y, z) = (0, 0, \cos(2\pi n x))$$

does not converges to zero. $\qquad\square$

*Remark* A.2. Note that for a $G_n$ set in the proof of the above theorem A.1, $\int_\Omega \|\nabla \times G_n\|^2 \, d\mathbf{x} = \frac{1}{2}$ is a positive constant independent of $n$. This implies that we can prevent the pathological example above by adding the curl-free loss term. Therefore, the curl-free term is necessary to accurately learn the gradient field $G$.

## A.2   Empirical Validation

To examine the practical effect of the curl-free term on learning a conservative vector field, we include experimental results on a simple example of a sphere of radius 0.5 centered at the origin. Figure 8 depicts the level set contours of the trained $u$ of a cross section cut at the planes $x = 0.2$ and $0.4$, along with the vector field of the trained $G$ projected onto this plane together with the gradient field of the true SDF. We note that the $p$-Poisson equation (1) gives an SDF that is positive on the interior of the surface and negative on the outside, however, the contours depicted in Figures 8 and 9 are of the opposite sign of the trained $u$. As shown in the Figure 8, the model trained without the curl-free term learns a vector field $G$ that is not curl-free, resulting in $G$ being distinct from the true gradient field. This ultimately impedes $u$ from correctly learning the SDF. On the other hand, it is evident that the model trained with the curl-free term converges fairly close to the true gradient field. This ultimately helps $u$ to accurately learn the SDF.

# B   Implementation Details

In this section, we provide more details about the implementation for reproducibility. Note that our code is built on top of IGR [2] (MIT License).

---

[2]`https://github.com/amosgropp/IGR`

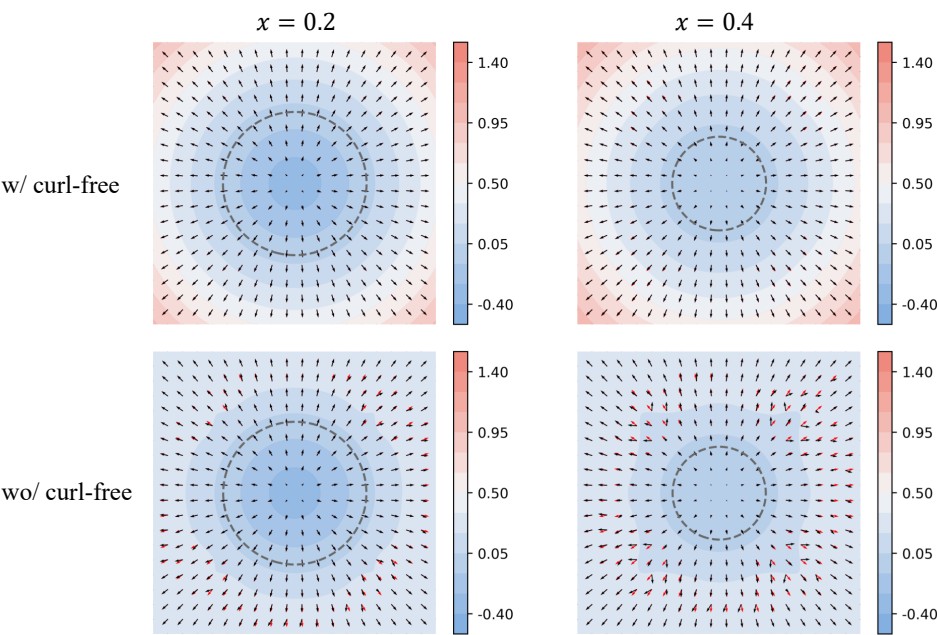

Figure 8: The trained results of a cross section cut in planes $x = 0.2$ (left) and $x = 0.4$ (right). The level-sets show the signed distance fields $u$ learned by the proposed model with (top) and without (bottom) the curl-free term. Dashed contours depict the learned zero level set. Quivers represent the vector field of the trained auxiliary variable $G$ and the true gradient fields are plotted in red arrows.

## B.1 Experimental Setup

**Parameter Tuning** The proposed training loss $\mathcal{L}_{\text{total}}$ (13) is a weighted sum of five loss terms with four regularization parameters $\lambda_1, \lambda_2, \lambda_3,$ and $\lambda_4$. In all surface reconstruction experiments, we use $\lambda_1 = 0.1$, $\lambda_2 = 0.0001$, $\lambda_3 = 0.0005$, and $\lambda_4 = 0.1$. In the proposed model, $p$ is also a hyperparameter to be chosen. Considering the theoretical fact that $p$ should be infinitely large and numerical simplicity, we set $p = \infty$. We empirically confirm no significant difference between when $p = 100$ and when $p = \infty$. Moreover, we set the smoothing parameter $\epsilon = 1$ for approximating Dirac delta in (13).

**Network Architecture** As in previous studies [44, 23, 37], we represent the primary and auxiliary outputs by a single 8-layered multi-layer perceptron (MLP) $\mathbb{R}^3 \to \mathbb{R}^7$ with 512 neurons and a skip connection to the fourth layer, but only the output dimension of the last layer is increased by six due to the two auxiliary variables; see Figure 2. We use softplus activation function $\alpha(x) = \frac{1}{\beta} \ln(1 + e^{\beta x})$ with $\beta = 100$. Network weights are initialized by the geometric initialization proposed in [3].

**Training details** The gradient and the curl of networks are computed with auto-differentiation library (`autograd`) [46]. In all experiments, we use the Adam optimizer [32] with learning rate $10^{-3}$ decayed by 0.99 every 2,000 iterations. At each iteration, we uniform randomly sample 16,384 points $\mathbf{x} \in \mathcal{X}$ from the point cloud $\mathcal{X}$. We sample the collocation points of $\Omega$ as provided in [23]. The collocation points consist of global points and local points. The local collocation points are sampled by perturbing each of the 16,384 points drawn from the point cloud with a zero mean Gaussian distribution with a standard deviation equal to the distance to the 50th nearest neighbor. The global collocation points are made up of approximately 2,000 points from the uniform distribution $U(-\eta, \eta)$ with $\eta = 1.1$. $F = \frac{1}{3}\mathbf{x}$ is utilized in all experiments.

**Baseline models**   For baseline models on the Thingi10K dataset, we use the official codes of IGR [2] (MIT License), SIREN[3] (MIT License), and DiGS [4] (MIT License). We faithfully follow the official implementation to train each model without normal prior. For the variable splitting representation of the eikonal equation (9), there is a single auxiliary output. Consequently, we use the same 8 layer MLP with 512 nodes, but a network with an output dimension of 4. We normalize the auxiliary output to make it a unit norm, and use the normalized one to represent $H$.

## B.2   Evaluation

**Metrics**   We measure the distance between two point clouds $\mathcal{X}$ and $\mathcal{Y}$ by using the standard one-sided and double-sided $\ell_1$ Chamfer distances $d_{\overrightarrow{C}}$, $d_C$ and Hausdorff distances $d_{\overrightarrow{H}}$, $d_H$. Each are defined as follows:

$$d_{\overrightarrow{C}}\left(\mathcal{X}, \mathcal{Y}\right) = \frac{1}{|\mathcal{X}|} \sum_{\mathbf{x} \in \mathcal{X}} \min_{\mathbf{y} \in \mathcal{Y}} \|\mathbf{x} - \mathbf{y}\|_2\,,$$

$$d_C\left(\mathcal{X}, \mathcal{Y}\right) = \frac{1}{2} \left( d_{\overrightarrow{C}}\left(\mathcal{X}, \mathcal{Y}\right) + d_{\overrightarrow{C}}\left(\mathcal{Y}, \mathcal{X}\right) \right)\,,$$

$$d_{\overrightarrow{H}}\left(\mathcal{X}, \mathcal{Y}\right) = \max_{\mathbf{x} \in \mathcal{X}} \min_{\mathbf{y} \in \mathcal{Y}} \|\mathbf{x} - \mathbf{y}\|_2\,,$$

$$d_H\left(\mathcal{X}, \mathcal{Y}\right) = \max \left\{ d_{\overrightarrow{H}}\left(\mathcal{X}, \mathcal{Y}\right) + d_{\overrightarrow{H}}\left(\mathcal{Y}, \mathcal{X}\right) \right\}\,.$$

When we estimate the distance from a surface, we sample $10M$ uniformly random points from the surface and then measure the distance from the sampled point clouds by the metrics defined above.

Furthermore, in order to measure the accuracy of the trained gradient field, we evaluate Normal Consistency (NC) [41] between the learned $G$ and the surface normal as follows: from given an oriented point cloud $\mathcal{X} = \{\mathbf{x}_i, \mathbf{n}_i\}_{i=1}^N$ comprising of sampled points $\mathbf{x}_i$ and the corresponding outward normal vectors $\mathbf{n}_i$, NC is defined by

$$NC\left(G, \mathbf{n}\right) = \frac{1}{N} \sum_{i=1}^N \left| G\left(\mathbf{x}_i\right)^{\mathrm{T}} \mathbf{n}_i \right|, \tag{17}$$

the average of the absolute dot product of the trained $G$ and the surface normals.

**Level set extraction**   We extract the zero level set of a trained neural network $u$ by using the classical marching cubes meshing algorithm [38] on a $512 \times 512 \times 512$ uniform grid.

## C   Additional Results

### C.1   Uniqueness of the solution of $p$-Poisson equation

In this section, we provide a numerical example supporting the strength of the proposed model regarding the uniqueness of the solution to the $p$-Poisson equation. The given points are located on a cube centerd at the origin with an edge of the length 1. We consider IGR [23] as an eikonal-rooted baseline, and we train IGR and the proposed PINC with the following three different network initializations: The geometric initialization [23] that IGR originally used and the Kaiming uniform initialization [26] with two different random seeds. The results are summarized in Figure 9. The results show that IGR converges to different solutions depending on the model initializations. In particular, IGR fails to learn the SDF of the cube except for the geometric initialization. On the other hand, the results of PINC with the same initializations show that the proposed model converges to the SDF in all three cases. The numerical results of the chosen example show that the proposed method can pursue the unique solution of the PDE.

### C.2   Additional comparison with models utilizing surface normals

In Section 4, we made a comparison with models that do not use the surface normal $\mathbf{n}$ as a supervision. Here, we additionally consider a comparison with models that leverage normal supervision. We

---

[3]`https://github.com/vsitzmann/siren`
[4]`https://github.com/Chumbyte/DiGS`

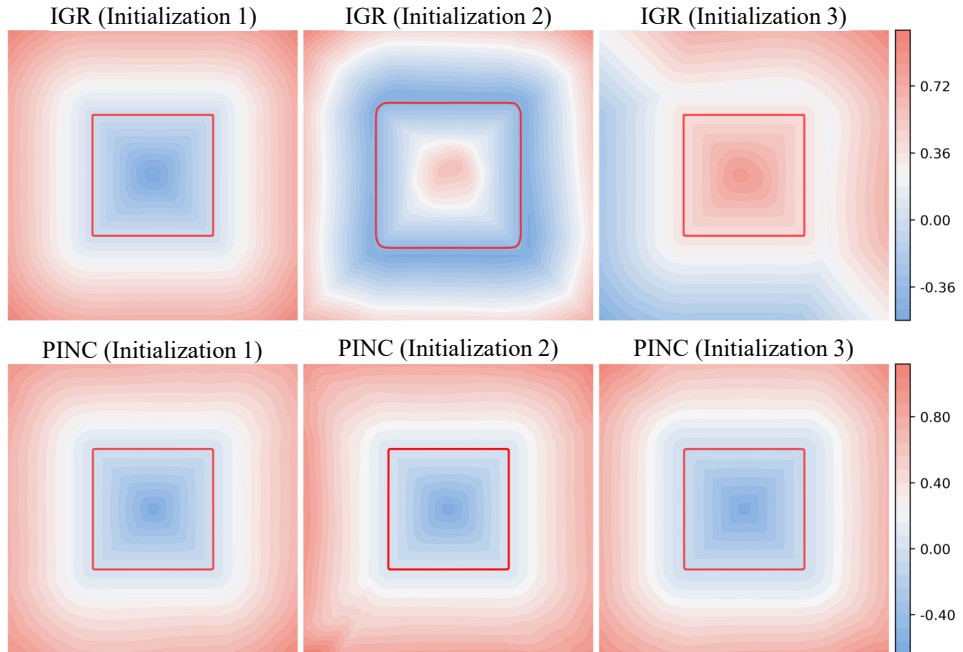

Figure 9: Experimental results show whether a method can find the SDF from different network initializations. IGR and PINC are trained on the synthetic cube data with three network initializations: geometric initialization (initialization 1) and Kaiming initialization with two different random seeds (initializations 2 and 3). Each depicts the trained level-set contours of a cross-section cut in the plane x = 0. Red contours depict the trained zero level set of numerical solutions.

Table 4: Comparison with models that use the surface normal supervision **n** on SRB. The proposed model PINC did not utilize the surface normal.

| | Model | Anchor GT $d_C$ | $d_H$ | Scans $d_{\overrightarrow{C}}$ | $d_{\overrightarrow{H}}$ | Daratech GT $d_C$ | $d_H$ | Scans $d_{\overrightarrow{C}}$ | $d_{\overrightarrow{H}}$ | DC GT $d_C$ | $d_H$ | Scans $d_{\overrightarrow{C}}$ | $d_{\overrightarrow{H}}$ | Gargoyle GT $d_C$ | $d_H$ | Scans $d_{\overrightarrow{C}}$ | $d_{\overrightarrow{H}}$ | Loard Quas GT $d_C$ | $d_H$ | Scans $d_{\overrightarrow{C}}$ | $d_{\overrightarrow{H}}$ |
|---|---|---|---|---|---|---|---|---|---|---|---|---|---|---|---|---|---|---|---|---|---|
| w/n | VisCO | 0.21 | 3.00 | 0.15 | 1.07 | 0.26 | 4.06 | 0.14 | 1.76 | 0.15 | 2.22 | 0.09 | 2.76 | 0.17 | 4.40 | 0.11 | 0.96 | 0.12 | 1.06 | 0.7 | 0.64 |
| | IGR | 0.22 | 4.71 | 0.12 | 1.32 | 0.25 | 4.01 | 0.08 | 1.59 | 0.17 | 2.22 | 0.09 | 2.61 | 0.16 | 3.52 | 0.06 | 0.81 | 0.12 | 1.17 | 0.07 | 0.98 |
| | SAP | 0.34 | 8.83 | 0.09 | 2.93 | 0.22 | 3.09 | 0.08 | 1.66 | 0.17 | 3.30 | 0.04 | 2.23 | 0.18 | 5.54 | 0.05 | 1.73 | 0.13 | 3.49 | 0.04 | 1.17 |
| wo/n | **PINC** | 0.29 | 7.54 | 0.09 | 1.20 | 0.37 | 7.24 | 0.11 | 1.88 | 0.14 | 2.56 | 0.04 | 2.73 | 0.16 | 4.78 | 0.05 | 0.80 | 0.10 | 0.92 | 0.04 | 0.67 |

consider three baseline models as follows: (i) IGR that evaluates using the surface normal, (ii) VisCo [49], a grid-based method based on the viscosity regularized eikonal equation, and (iii) Shape As Points (SAP) [48], a model that revisits the classical Poisson Surface Reconstruction (PSR) [31] using deep learning. The results are reported in Table 4. We can see that the proposed model performs on par with baselines, despite not utilizing the surface normal. Considering that all of the baselines are PDE-based INR models, the results exhibit the effectiveness of the proposed model (11) in reconstructing a surface from the sole use of raw point clouds.

It is worth note that the proposed model may be interpreted as PSR because of (8). More precisely, the Euler-Lagrange equation of (8) says that the variational problem (8) for finding a scalar function $u$ whose gradient best approximates a given vector field $G$ transforms into the following Poisson problem:

$$\begin{cases} \triangle u = \nabla \cdot G & \text{in } \Omega \\ u = 0 & \text{on } \Gamma. \end{cases} \tag{18}$$

In the conventional PSR, the gradient field is set to surface normals. Thus, the auxiliary variable $G$ can be regarded as playing a role of surface normals for PSR. However, the vector field $G$ in the proposed model is not obtained from the oriented point cloud, but the learnable function that is trained with $u$ at the same time. Moreover, since we bake the $p$-Poisson equation into $G$ as a hard constraint in (7), we obtain a continuous SDF rather than an indicator function like PSR and SAP. The results confirm that simultaneous training of the gradient field and the SDF, that is, the variable

Table 5: Normal consistency of reconstructed surfaces on SRB.

| Model | Anchor | Daratech | DC | Gargoyle | Lord Quas |
|-------|--------|----------|-----|----------|-----------|
| IGR | 0.9706 | 0.8526 | 0.9800 | 0.9765 | 0.9901 |
| SIREN | 0.9438 | **0.9682** | 0.9735 | 0.9392 | 0.9762 |
| DiGS | **0.9767** | 0.9680 | 0.9826 | 0.9788 | 0.9907 |
| SAP | 0.9750 | 0.9414 | 0.9636 | 0.9731 | 0.9838 |
| **PINC** | 0.9754 | 0.9311 | **0.9828** | **0.9803** | **0.9915** |

Table 6: Normal consistency of reconstructed surfaces on Thingi10K.

| Model | Squirrel | Pumpkin | Frogrock | Screstar | Buser head |
|-------|----------|---------|----------|----------|------------|
| IGR | **0.9820** | 0.9565 | 0.9509 | 0.9709 | 0.9249 |
| SIREN | 0.9529 | 0.8996 | 0.9035 | 0.9142 | 0.8860 |
| DiGS | 0.9557 | 0.9353 | 0.9468 | 0.9386 | 0.9171 |
| SAP | 0.9791 | 0.9520 | 0.9319 | 0.9767 | 0.9004 |
| **PINC** | 0.9816 | **0.9583** | **0.9545** | **0.9805** | **0.9376** |

splitting method, achieves similar or even better surface restoration than SAP, even without using the given surface normal **n**.

## C.3 Additional quantitative results

We reported Chamfer distances and Hausdorff distances in Tables 1 and 2, but these two metrics do not reflect the complete quality of the restored surface. Here, we evaluate Normal Consistency (NC) (17) which measures how well the model can capture higher order information of the surface. The results on both SRB and Thingi10K datasets are summarized in Tables 5 and 6, respectively. Overall, the proposed model achieves a better NC score than baseline models. In particular, the results show that the proposed model achieves superior NC for the tested examples than SAP, even though it does not employ surface normal supervision.

Moreover, we measure Chamfer distance and Hausdorff distance for ablation studies reported in Figures 6 and 7 of Section 4.3 and summarized them in the Tables 7 and 8, respectively.

**More results on effect of** $p$ Theoretically, an accurate SDF can be obtained as $p$ grows infinitely. That the same story continues in practice is confirmed by the results shown in Figure 10. We can see that the larger $p$ induces a better reconstruction. This phenomenon is also observed in Figure 7. Moreover, it can be seen that $p = \infty$, which we used in the implementation, gives a similar qualitative result to $p = 100$. These experimental results once again remind us how important it is to be able to use a large $p$.

Furthermore, we provide numerical verification for the use of $p = \infty$ in Figure 11. For notational convenience, we use the subscript $u_p$ to denote the dependence of the solution on the parameter $p$. Figure 11 depicts graphs of the mean squared error (MSE) of $u_p$ and $u_\infty$ over different $p$. MSEs are computed by discretizing the computational domain $\Omega$ into a $100 \times 100 \times 100$ uniform grid. The results show that the MSE decreases as $p$ increases. In other words, it confirms that $u_p$ is getting closer to $u_\infty$ as $p$ grows, which supports the justification for using $p = \infty$.

Table 7: Quantitative results on the effect of area loss on daratech.

| Model | GT | | Scans | |
|-------|-------|-------|-------|-------|
| | $d_C$ | $d_H$ | $d_{\overrightarrow{C}}$ | $d_{\overrightarrow{H}}$ |
| PINC wo/ area loss | 4.26 | 53.34 | 0.20 | 2.81 |
| PINC w/ area loss | 0.37 | 7.24 | 0.11 | 1.88 |

Table 8: Quantitative results on the ablation study of $p$.

| | Gargoyle | | | | Anchor | | | |
| | GT | | Scans | | GT | | Scans | |
| Model | $d_C$ | $d_H$ | $d_{\vec{C}}$ | $d_{\vec{H}}$ | $d_C$ | $d_H$ | $d_{\vec{C}}$ | $d_{\vec{H}}$ |
|---|---|---|---|---|---|---|---|---|
| $p = 2$ | 3.96 | 43.13 | 0.51 | 6.31 | 4.32 | 46.66 | 0.77 | 14.58 |
| $p = 10$ | 0.22 | 8.14 | 0.10 | 1.19 | 0.50 | 7.24 | 0.12 | 3.02 |
| $p = 100$ | 0.17 | 4.90 | 0.10 | 0.82 | 0.31 | 7.20 | 0.13 | 1.80 |
| $p = \infty$ | 0.16 | 4.78 | 0.05 | 0.80 | 0.29 | 7.19 | 0.11 | 1.17 |

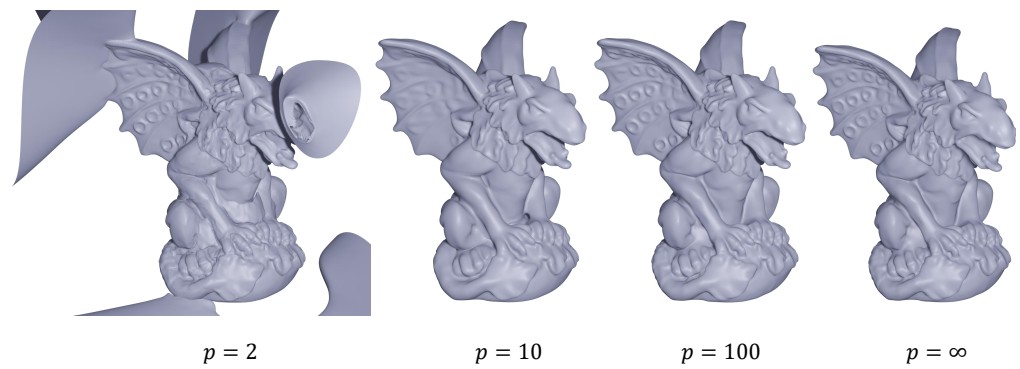

$p = 2$      $p = 10$      $p = 100$      $p = \infty$

Figure 10: Quality of surface reconstruction with varying $p$ from $p = 2$ to $p = \infty$.

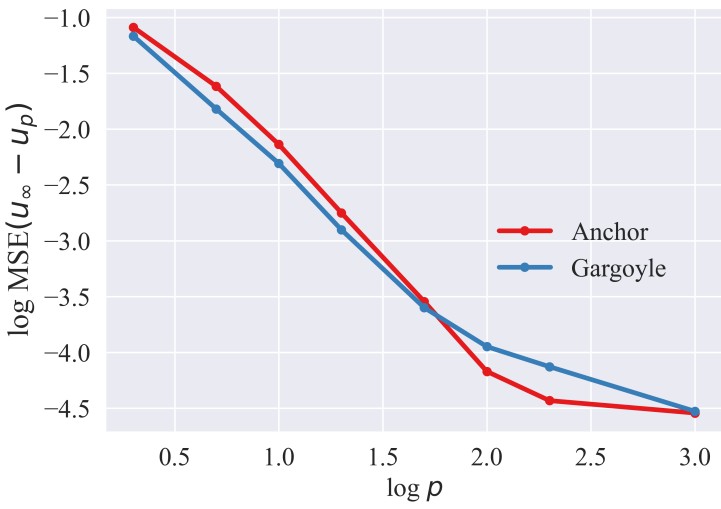

Figure 11: MSEs of $u_p$ and $u_\infty$ over different $p$.

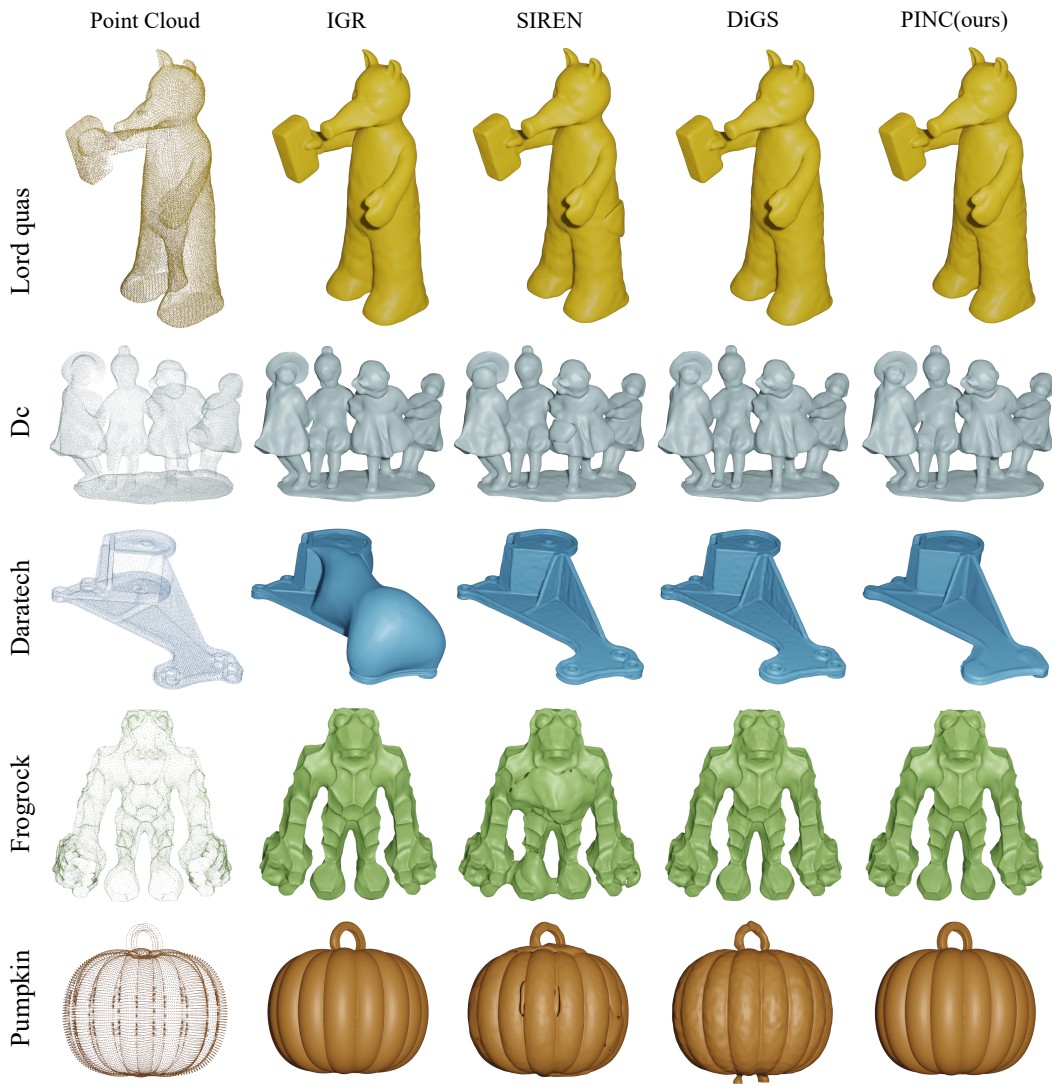

Figure 12: Additional qualitative results of the surface reconstruction on SRB and Thingi10K datasets.

## C.4 Additional qualitative results

Figure 12 provides additional qualitative results of surface reconstruction on SRB and Thingi10K discussed in Section 4.1.

**Reconstruction of large point clouds** We further provide qualitative results for surface reconstruction from large models taken from Thingi10K. The adopted point clouds consist of from 35K to 980K vertices. Figure 13 depicts the qualitative reconstruction results of PINC on these large point clouds. The model is trained with the same configuration used in Section 4.1.

## C.5 Training/Inference time

To investigate the computational time of the proposed model, We carefully measured average execution time compared to baselines. In the Table 9, we report the average training time per iteration and inference time at a resolution of $32^3$ voxels. The proposed model requires more computational cost than baseline models because of the computation on curl using automatic differentiation.

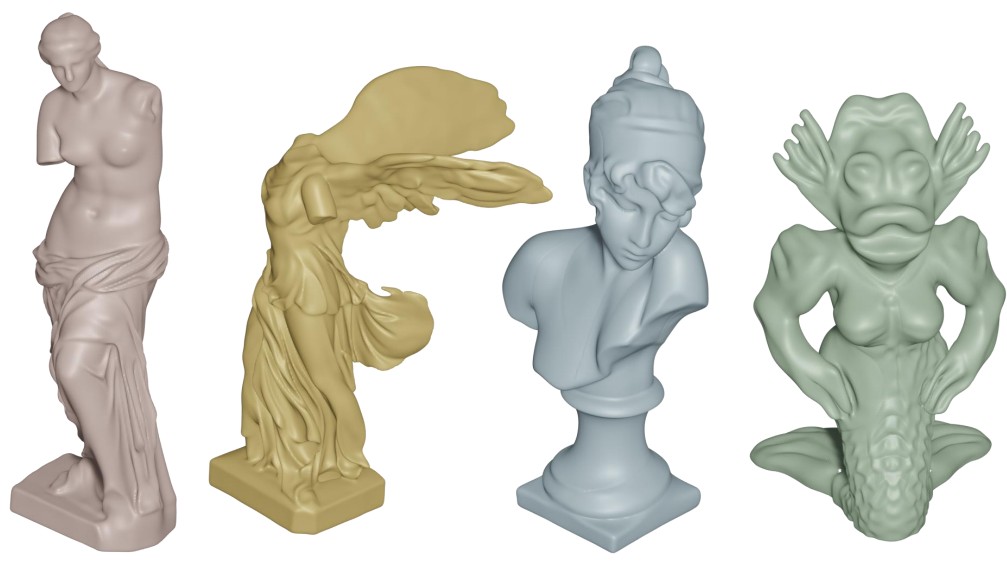

Figure 13: Reconstructed surfaces of large point clouds from Thingi10K.

Table 9: Training and inference times for surface reconstruction on SRB.

| Model | IGR | SIREN | DiGS | **PINC** |
|---|---|---|---|---|
| Training time (ms/iteration) | 48.34 | 13.11 | 52.34 | 295.01 |
| Inference time (ms) | 6.86 | 3.51 | 4.39 | 6.93 |

