# OpenReview forum: "$p$-Poisson surface reconstruction in curl-free flow from point clouds"
_NeurIPS.cc/2023/Conference — NeurIPS 2023 poster_

### Official Review · Reviewer_2pnE · 2023-07-02

**Soundness:** 2 fair
**Presentation:** 3 good
**Contribution:** 3 good
**Rating:** 7
**Confidence:** 5

**Summary:**

This content seems interesting. I like that the authors gave considerable thought to improving surface reconstruction from the perspective of vector field processing. The paper targets two challenging problems in surface reconstruction: (1) removing the requirement for surface normal, and (2) improving the over-smoothness. However, there are technical details that need to be addressed before I can accept it.

**Strengths:**

(1) The problems are challenging.
(2) Methodology is novel.
(3) Improvements are observed.


**Weaknesses:**

In a p-Poisson equation, you have the p-Laplacian instead of the Laplacian. This is equivalent to giving weights to the gradient of an implicit function. And this weight is based on the magnitude of the gradient itself. The geometric intuition behind this weight is not clearly stated.

**Questions:**

major:

(1) Add a comparison for the learned vector field. From equation (8), your G plays the role of the vector field derived from the oriented point cloud in PoissonRecon. It might be beneficial to compare G with the vector field formulated by surface normal.

(2) In section 3.2, you mentioned the goal is to progressively enlarge lambda to infinity. If so, have you tried getting rid of the first term of equation (8)? I don’t think the method will fall apart. You would only be downgrading from ScreenedPoissonRecon to PoissonRecon.

(3) The result in Figure 3 is a bit confusing. What’s your sampling strategy? Are you sampling all the vertices from the mesh? Because from the Screwstar, it seems you are not sampling uniformly. Could that partially be the reason the SIREN has a broken surface? Can you try to uniformly sample the star and see if there’s any improvement in the completeness?

(4) The result in Figure 7 (a) had some clear artifacts. Why did that happen? When p = 2, your energy looks almost the same as ScreenedPoissonRecon. However, when I ran ScreenedPoissonRecon on this exact model, I did not see this artifact.

minor:

(1) The statement “irrotational flow” only appeared once in the title. If you want to use the statement, you need to be consistent in the writing. At least mention and explain it in the introduction.

(2) In Chapter 4.3, you used the subtitle “Effect of curl-free constraint” twice. I believe you meant to say “Effect of minimal surface area constraint” for the second one.

**Limitations:**

yes

---

> ### Author Rebuttal · Authors · 2023-08-09
>
> We would like to thank the reviewer for the constructive comments. Below, we carefully address the reviewer's comments:
>
> **Q1. In a p-Poisson equation, you have the p-Laplacian instead of the Laplacian..**
>
> **Reply**: In the $p$-Poisson equation $-\triangle_p u = 1$, by letting $p$ in the weight $\parallel\nabla u \parallel^{p-2}$ grows infinitely, we can obtain the SDF without normal supervision. More precisely, the $p$-Laplacian $\triangle_p$ can be decomposed as follows (Kawohl, 2016):
> $$\frac{1}{p}\triangle_p u= \frac{1}{p}\parallel\nabla u \parallel^{p-1}\triangle_1 u+ \frac{p-1}{p}\parallel\nabla u \parallel^{p-2}\triangle_\infty u,$$
> where $\triangle_1 u = \nabla \cdot \left( \frac{\nabla u}{\parallel\nabla u\parallel} \right)$ geometrically represents the mean curvature of the isosurfaces of $u$ and $\triangle_\infty u = \nabla u^T H\nabla u$ stands for the second derivative in the steepest ascending direction. Here, $H$ denotes the Hessian matrix of $u$. The parameter $p$ directly represents the weights between these two terms. As $p$ becomes larger, the weight of the second part is larger and eventually converges to the infinite Laplacian $\triangle_\infty u$. On the other hand, if we set $p=2$ (the Laplacian), we couldn't get the SDF.
>
> **Q2. Add a comparison for the learned vector field..**
>
> **Reply**: We agree that $G$ plays the role of normal field in PoissonRecon because we find a scalar function $u$ whose gradient best fit the vector field $G$ by minimizing $\int_\Omega \left\Vert \nabla u - G\right\Vert^2$. Following the reviewer's recommendation, we measure the difference between the given surface normal $n$ and the learned $G$ on SRB dataset. Given an oriented point cloud together with surface normals
> $\lbrace x_i, n_i \rbrace$, $i=1,\cdots,N$, we measure the cosine similarity $G^Tn\coloneqq\frac{1}{N}\sum_{i=1}^N | G\left(x_i\right)^T n_i |$. In addition, we also report the cosine similarity of the gradient field of learned SDF $u$ and the surface normal. Results are reported in the Table below. The results confirm that $G$ is accurately trained.
>
> |Model|Anchor|Daratech|DC|Gargoyle|Lord Quas|
> |:---:|:---:|:---:|:---:|:---:|:---:|
> |$G^Tn$|0.9868|0.9544|0.9941|0.9924|0.9965|
> |$\nabla u^Tn$|0.9870|0.9579|0.9946|0.9929|0.9966|
>
>
> **Q3. In section 3.2, you mentioned the goal is to progressively enlarge lambda to infinity..**
>
> **Reply**: In the standard penalty method, using a progressively larger value of the penalty parameter is theoretically necessary to obtain a solution of the constrained optimization. In the numerical simulation, we could make a procedural sequential approach to increase $\lambda_1$ in (8). However, in practice, we cannot make the value of $\lambda_1$ infinitely large (the line number 147 of the paper). As $\lambda_1 \rightarrow \infty$, the balancing between $\parallel \nabla u - G\parallel$ and $\parallel \nabla \times \tilde{G}\parallel$ is going to be broken and then it is difficult to obtain the curl-free constraint which is one of crucial parts of the proposed method; please check Q3 of the author response to the reviewer rqMu for the necessity of the curl-free term. Moreover, without the first term, the solution is up to constant and then there should be an extra step to find a unique solution. Since the first term is the fidelity working like Dirichlet boundary condition, it enforces the implicit function should be zero at the point cloud.
>
> **Q4. The result in Figure 3 is a bit confusing..**
>
> **Reply**: When we trained SIREN, points on $\Gamma$ were uniform randomly sampled from the original point cloud and we set collocation points of $\Omega$ as the uniform grid. The reason for SIREN to restore the broken surface seems to be the choice of activation function and network initialization. We can see from Figure 3 that IGR has smooth surfaces, but the difference between SIREN and IGR is the choice of activation function and network initialization. IGR uses the softplus activation function and initializes the network to be the SDF for the sphere. SIREN, on the other hand, adopts the sine function as the activation and an initialization that preserves the distribution of activations through its layers.
>
> **Q5. The result in Figure 7 (a) had some clear artifacts..**
>
> **Reply**: In the proposed model, there are a few differences with ScreenedPoissonRecon (SPR). Firstly, the corresponding Euler-Lagrange (EL) equation of the proposed objective is not reduced to the screened Poisson equation because the first loss term considers $\left\vert u\right\vert$ rather than $\left\vert u\right\vert^2$ and the integration is computed over $\Gamma$ rather than the whole computational domain $\Omega$. The first term in (8) corresponds to the boundary condition. Secondly, $G$ is not the vector field obtained from the oriented point cloud, but the learnable function that is simultaneously trained with $u$. Moreover, since we impose the $p$-Poisson equation on $G$ as a hard constraint (7), we obtain an SDF rather than an indicator function like SPR. When we use small $p$ in the view of finding the SDF, the construction of $G$ leads to degraded results; see Figure 7. When $p=2$, the optimal solution $u$ is the following PDE:
> $$\triangle u = \nabla\cdot G= -1$$
>
> with Dirichlet boundary condition $u=0$ on $\Gamma$, which is far away from the SDF.
>
> **Q6. The statement “irrotational flow” only appeared once in the title..**
>
> **Reply**: We completely agree with the reviewer. We will revise the manuscript as recommended.
>
> **Q7. In Chapter 4.3, you used the subtitle “Effect of curl-free constraint” twice..**
>
> **Reply**: Thank you for pointing this out. We will revise the manuscript as recommended.
>
> **Reference**
>
> B. Kawohl et al. On the geometry of the $ p $-Laplacian operator. arXiv preprint arXiv:1604.07675, 2016.

---

> > ### Comment · Reviewer_2pnE · 2023-08-13
> >
> > The rebuttal has been thorough. I recommend authors add additional explanation and try to be as clear as possible to convey the motivation/intuition of adopting p-Poisson equation in the final manuscript. Regardless, this work has enough novelty. I change my score to 7, and recommend acceptance of this paper.

---

### Official Review · Reviewer_WkvV · 2023-07-05

**Soundness:** 1 poor
**Presentation:** 1 poor
**Contribution:** 3 good
**Rating:** 5
**Confidence:** 5

**Summary:**

The paper introduces an intriguing approach to surface reconstruction, but further enhancements in terms of completeness and evaluations would strengthen its contribution to the field. Also, the gradient of the SDF acts as an auxiliary network output and incorporated the Poisson equation as a hard constraint. The proposed method was also used to propose a more accurate representation. They perform some experiments on standard benchmark datasets to demonstrate superior and robust reconstruction. In my view, the proposed method cannot achieve the best one numerically on average (table 1).

**Strengths:**

This paper proposes a novel surface reconstruction method using the p-Poisson equation and a curl-free constraint, which is highly interesting. It demonstrates superior performance compared to previous works.

**Weaknesses:**

The authors have submitted a revision of the full paper in the supplementary material, it may be considered a violation of the rules. It may be appropriate to consider resubmitting the paper to another venue due to the violation.

There are several incomplete pieces of information.

It would be beneficial to include evaluations to measure surface quality, such as normal consistency, as well as provide details on training and inference times.

**Questions:**

1. Your method appears to be computationally intensive. Can you provide information on the training and inference times compared to other methods?

2. While your results look promising, including quantitative results on surface quality would be valuable.

3. It would be unfair to directly compare the proposed method with approaches that utilize normal information. However, conducting experiments and comparisons with methods such as "shape as points" from the URL [https://pengsongyou.github.io/sap] would provide valuable insights and contribute to a more comprehensive evaluation of the proposed method.

4. What are lines 457-460?

**Limitations:**

The limitation has been roughly discussed in the paper.
Poisson-based methods, including the proposed approach, are unable to handle open surfaces.

---

> ### Author Rebuttal · Authors · 2023-08-09
>
> We would like to thank the reviewer for the thoughtful comments. Below, we carefully address the reviewer's comments:
>
> **Q1. The authors have submitted a revision of the full paper in the supplementary material..**
>
> **Reply**: The reviewer's point is indeed correct if the main argument, main idea, or computational results of the proposed algorithm had been changed between the original paper and the paper with the supplementary material. The paper with the supplementary material has precisely the same arguments and ideas as the original paper. The only changed parts are i) the results of SIREN, ii) a typographical error in Figure 7, and iii) some of the incomplete information in the reference. The superiority of the results remains almost the same, so none of the main arguments in the paper are affected. We would like to note that we revised the manuscript to provide accurate values for the reviewer's better judgment, not because we wanted to change or strengthen the main argument.
>
> **Q2. It would be beneficial to include evaluations to measure surface quality,..**
>
> **Reply**: Thanks for the comments on further evaluations. We provide the related answers in Q4 for a measure of surface quality regarding surface quality; see Tables 2 and 3. Training/inference times are reported in the answer to Q3; see Table 1.
>
> **Q3. Your method appears to be computationally intensive..**
>
> **Reply**: We investigate the training/inference times of the proposed model compared to other models. In the Table 1 below, we report the average training time per iteration on SRB dataset and inference time at a resolution of $32^3$ voxels.
> As the reviewer mentioned, the proposed model requires more computational cost than baseline models because of the computation on curl using automatic differentiation.
>
> [Table 1] Training/Inference times
> |Time|IGR|SIREN|DiGS|PINC (ours)|
> |:---|:---:|:---:|:---:|:---:|
> |Training time (ms/iteration)|48.34| 13.11 | 52.34 | 295.0|
> |Inference time (ms)| 6.86 | 3.51 | 4.39 | 6.93|
>
> **Q4. While your results look promising, including quantitative results on surface quality would be valuable.**
>
> **Reply**: Thank you for the suggestion. As normal consistency (L. Mescheder, 2019) is recommended in Q2, it is tested and the results are reported in the Tables 2 and 3 below. Overall, compared to other models, the proposed model achieves a better normal consistency.
>
> [Table 2] Normal Consistency on SRB
> |Model|Anchor|Daratech|DC|Gargoyle|Lord Quas|
> |:---:|:---:|:---:|:---:|:---:|:---:|
> |IGR|0.9706|0.8526|0.9800|0.9765|0.9901|
> |SIREN|0.9438|**0.9682**|0.9735|0.9392|0.9762|
> |DiGS|**0.9767**|0.9680|0.9826|0.9788|0.9907|
> |SAP|0.9750|0.9414|0.9636|0.9731|0.9838|
> |PINC (ours)|0.9754|0.9311|**0.9828**|**0.9803**|**0.9915**|
>
>
> [Table 3] Normal Consistency on Thingi10K
> |Model|Squirrel|Pumpkiin|Frogrock|Scrwstar|Buser head|
> |:---:|:---:|:---:|:---:|:---:|:---:|
> |IGR|**0.9820**|0.9565|0.9509|0.9709|0.9249|
> |SIREN|0.9529|0.8996|0.9035|0.9142|0.8860|
> |DiGS|0.9557|0.9353|0.9468|0.9386|0.9171|
> |SAP|0.9791|0.9520|0.9319|0.9767|0.9004|
> |PINC (ours)|0.9816|**0.9583**|**0.9545**|**0.9805**|**0.9376**|
>
> **Q5. It would be unfair to directly compare the proposed method with approaches that utilize normal information..**
>
> **Reply**:  Thank you for the comment. Following the reviewer's suggestion, we include the comparison with Shape As Points (SAP) on both SRB and Thingi10K dataset using three metrics for quantitative evaluation: Chamfer distance (CD) and Hausdorff distance (HD) are summarized in Tables A5 and A6 in the attachment, and evaluation of normal consistency (NC) is reported in Tables 2 and 3 in the response to Q4. The results show that the CD and HD of the proposed model are similar to SAP, despite not utilizing the given surface normal. Furthermore, the proposed model, which learns the gradient field of the $p$-Poisson equation instead of using the given surface normal, achieves a better overall NC.
>
> SAP is based on Poisson Surface Reconstruction (PSR; Kazhdan, 2006). The proposed model may be interpreted as PSR because of (8). However, the vector field $G$ in the proposed model is not obtained from the oriented point cloud, but the learnable function that is trained with $u$ at the same time. Moreover, since we bake the $p$-Poisson equation into $G$ as a hard constraint in (7), we obtain a continuous SDF rather than an indicator function like PSR and SAP. The results confirm that simultaneous training of the gradient field and the SDF, that is, the variable splitting method, achieves similar or even better surface restoration than SAP, even without using the given surface normal.
>
> **Q6. What are lines 457-460?**
>
> **Reply**: Thank you for pointing out the mistake. We will make sure that it will be deleted in the revised version.
>
> **References**
>
> L. Mescheder et al. Occupancy networks: Learning 3d reconstruction in function space. IEEE/CVF, 2019.
>
> M. Kazhdan et al. Poisson surface reconstruction. In Proceedings of the fourth Eurographics symposium on Geometry processing, 2006.

---

> > ### Comment · Reviewer_WkvV · 2023-08-16
> > **Reply**
> >
> > After reading the other reviews and your responses, I think some of the concerns are addressed well. Here, more comprehensive numerical evaluations are provided to demonstrate the quantitative performance of the proposed methods, I suggest the author add these experiments to the revised paper, including the numerical evaluations, training/inference time, and discussion with SAP. All of the responses have addressed my major concerns, instead of the unfair submission. After that, I am positive about the submission and will change my score to accept when I ignore the unfair submission. I have raised the issues to the ac and sac, and I have no other comments if it is uncritical for the submission.

---

### Official Review · Reviewer_FaEi · 2023-07-06

**Soundness:** 3 good
**Presentation:** 3 good
**Contribution:** 3 good
**Rating:** 7
**Confidence:** 4

**Summary:**

This paper considers the problem of reconstruction of a smooth surface from an unorganised point cloud. The proposed approach is based on neural implicit function while without normal information. The main contribution of this work is that they demonstrate that proper supervision of partial differentiable equation and fundamental properties of differential vector fields are enough to reconstruct high-quality surfaces. A novel part is to develop a variable splitting structure by introducing a gradient of the SDF as an auxiliary variable and a curl-free constraint on the auxiliary variable.

The experimental results somehow demonstrate the effectiveness on some aspects.

**Strengths:**

I like the idea by introducing auxiliary variable to solve the optimization problem under the framework of neural implicit function. Actually, they all both solve optimization problem. Therefore, the optimization strategy in numerical algorithm can be adopted for neural implicit function based representation. This paper shows a good example in this aspect, and might inspire some interesting along this direction.

**Weaknesses:**

The results shown in Tab. 1 and Tab. 2 are not good enough. Can you explain why the performance is not good enough for different dataset and metrics?

**Questions:**

With additional auxiliary variable, does the optimization take long time to converge? If yes, please list the computation time with more details.

**Limitations:**

As listed in the above.

---

> ### Author Rebuttal · Authors · 2023-08-09
>
> We would like to thank the reviewer for the valuable comments. Below, we carefully address the reviewer's comments:
>
> **Q1. The results shown in Tab. 1 and Tab. 2 are not good enough..**
>
> **Reply**: Each shape of data has a different level of difficulty due to its own challenging characteristics, such as complex topology, missing data, sharp corners, a high level of detail, and so on.
> Therefore, it is natural for the model to perform differently on these different data. The proposed model has the advantage of learning an SDF based on the $p$-Poisson equation, which implicitly represents the surface, allowing for accurate and smooth reconstruction of closed surfaces. Therefore, it is difficult to say that surface reconstruction of the proposed model performs better than other models on data that cannot highlight these advantages. We only confirmed that the experimental results of the proposed model achieved comparable results to leading INR models across the various data. Moreover, different metrics quantify different features.
> The proposed model seems to show better results with Chamfer distances than with Hausdorff distances.
> However, these two metrics do not reflect the complete quality of the restored surface. We evaluate normal consistency (L. Mescheder, 2019) as recommended by the reviewer WkvV (Please see Tables 2 and 3 below the answer to Q4). We would like to note that the proposed model achieves better normal consistency for the tested examples.
>
> **Q2. With additional auxiliary variable, does the optimization take long time to converge? If yes, ..**
>
> **Reply**: We agree with the reviewer that, in general, the computational cost of using and not using auxiliary variables is undoubtedly not the same. However, we have technical difficulties to make a fair comparison on this issue. In the case of auxiliary variable $\tilde{G}$, the results show a significant difference in performance with and without using it; see more details in Figure 5. It means that the convergent point may not be the same with and without using $\tilde{G}$, so it is difficult to compare convergence speeds under equivalent conditions when the exact solution is not specifically known. In the case of auxiliary variable $G$ cannot be excluded due to the construction of the proposed model. Nevertheless, analyzing the convergence speed or computational cost of using and not using auxiliary variables is indeed a crucial topic and it should be studied rigorously and mathematically with a very simple and meaningful loss function. We thank the reviewer for pointing out such a worthwhile future research topic. We will add this as future work in Section 5.

---

### Official Review · Reviewer_ymPA · 2023-07-06

**Soundness:** 3 good
**Presentation:** 3 good
**Contribution:** 3 good
**Rating:** 6
**Confidence:** 2

**Summary:**

The paper presents a surface reconstruction method that uses only raw point clouds. It enforces the Poisson surface equation implicitly over the SDF representation of the surface. As a consequence, it obtains smooth surfaces with preserved details without any 3D supervision or apriori knowledge of normals. The experiments show that the performance is comparable to sota methods which require data beyond raw point clouds.


**Strengths:**

The use of p-poisson equation to describe SDF is well-motivated.
The use of auxiliary variables relating to the gradient and curl of SDF is interesting and convincingly reduces the computational complexity.
Ablation study is a plus.
The performance is comparable to the methods that use either 3D supervision or oriented normals.
The performance on noisy data is generally better than sota. I think it is due to the fact the method uses only raw point clouds that serves as a benefit here. Normal computation on noisy point clouds can be disproportionately erroneous.

**Weaknesses:**

There is no theoretical motivation/argument provided to choose p-poisson over eikonal equation to describe SDFs.

IGR[23] can perform surface reconstruction without normals as well. It is not clear whether the authors used the Normals while evaluating IGR. A comparison with both: IGR with and without normals should have been considered.



**Questions:**

see weakness section

**Limitations:**

to some extent, the limitations are discussed.

---

> ### Author Rebuttal · Authors · 2023-08-09
>
> We thank the reviewer for the valuable feedback. We carefully address your questions as follows:
>
> **Q.1 There is no theoretical motivation/argument provided to choose p-poisson over eikonal equation to describe SDFs.**
>
> **Reply**: For a detailed response to the reviewer's question about the theoretical motivation for adopting the $p$-Poisson equation instead of the eikonal equation to describe SDFs, please check the response to the common question above.
>
> **Q2.IGR[23] can perform surface reconstruction without normals as well..**
>
> **Reply**: We compared the performance with IGR without normal vectors $n$. As the reviewer recommended, we present the comparison of results with IGR with and without $n$, and the proposed model in the Table A5 of the attachment.

---

> > ### Comment · Reviewer_ymPA · 2023-08-18
> >
> > Thank you for the rebuttal. It addresses most of my concerns. I am going to maintain my rating.

---

### Official Review · Reviewer_rqMu · 2023-07-07

**Soundness:** 3 good
**Presentation:** 2 fair
**Contribution:** 3 good
**Rating:** 6
**Confidence:** 4

**Summary:**

The paper considers the task of surface reconstruction from point clouds without normals with INRs. They consider solving for the SDF by solving the p-Poisson equation (with manifold constraint) as p\to\infty. However as the obvious loss function form of this is difficult to optimise, they consider a variable splitting strategy. They also do this w.r.t. ensuring that their gradient solution is conservative. Finally they add a minimal surface area regularisation term.

**Strengths:**

- Variable splitting is a nice approach to the issues with neural networks and automatic differentiation
- Model is backed strongly by theoretical intuition
- Auxilary variables sharing the network structure is nice
- Decent results

**Weaknesses:**

- It would be nice to have some intuition as to why you propose to use the p-Poisson equation rather than other PDEs like the Eikonal equation. At first glance it seems that the reason is because it is possible to describe it as a variational problem as shown in equation 2, however that doesn't get used by your method. Another argument you posit is that without the vanishing viscosity method a normal eikonal PDE based solution may produce a non-unique weak solution, why is it clear that your method does not produce a non-unique weak solution? Is it because of the curl-free constraint being enforced?
- How important is the enforcement of the loose eikonal constraint within the curl-free constraint? It somewhat diminishes the story of trying to solve the SDF problem using a different PDE to the eikonal equation.
- It is not clear why the curl-free constraint is needed. $\nabla u$ is curl-free by design, so isn't minimising $||\nabla u - G||$ in (8) sufficient? Why is a separate auxiliary variable necessary, apart from enforcing a loose Eikonal constraint by construction? Doesn't the argument about needed to set $\lambda_1$ infinitely large to enforce the constraint apply to $\lambda_3$ as well?
- The qualitative diagrams (Figure 5-7) for the ablation study are great for intuition and understanding, however you should have quantitative results on what happens when you remove each of those components (especially for curl-free)

I like this type of approach, willing to increase the score if clarity on the necessity of both the curl-free constraint and the eikonal construction constraint as well as quantitative ablations are given.

**Questions:**

- What is $F$? Is it kept to the example given, $1/3x$?
- I am confused why the curl-free constraint needed. By equation 10, the new curl-free constraint ensures that $G=\nabla v$ for some $v$, but why allow it to be some $v$ that is not the $u$ being outputed as the INR value, and/or instead constrain it to be similar to the current INR value $u(x)$?
- I would like more intuition on the role of the curl-free constraint. Figure 5 seems to show that it forces the model to pay attention to detail more, however I don't see why this is the case theoretically. Is it because the eikonal term is loosely being enforced within the curl-free objective by construction?
- The good results of IGR on Thingi10K seems unlikely given its bad performance on SRB, it is almost as good as your model. Is there a property of Thingi10K that causes this? Is there a reason your model and IGR would be so similar on Thingi10K?
- The performance of your model on Daratech in SRB seems a bit confusing, in the results it does really badly on $d_C$ however it looks fairly good in Figure 6?

**Limitations:**

Some discussion given.
No clear potential negative societal impact or broader societal impacts to discuss

---

> ### Author Rebuttal · Authors · 2023-08-09
>
> We thank the reviewer for the thoughtful feedback. We hereby carefully address your questions as follows:
>
> **Q1. It would be nice to have some intuition as to why..**
>
> **Reply**:  Please check the answer to the common question above.
>
> **Q2. How important is the enforcement of the loose eikonal constraint..**
>
> **Reply**: Imposing the curl-free condition helps to learn $G$ and $u$ accurately. The necessity of the curl-free constraint is explained in detail in the answer to Q3.
>
> **Q3. It is not clear why the curl-free constraint is needed..**
>
> **Reply**: Minimizing $L_2=\int_\Omega \parallel \nabla u_n - G_n\parallel^2 dx$ in (8) is not a pointwise-manner. There is a sequence {$u_n,G_n$} such that $L_2 \rightarrow 0$ but $G_n$ does not converge to a curl-free field. For every $u_n$ defined on $\Omega=[0,1]^3$, set $G_n(x,y,z)=\nabla u_n(x,y,z) + (0,1/n\sin(2\pi nx),0)$. Then, $L_2\rightarrow 0$ but $\nabla\times G_n=(0,0,\cos(2\pi nx))\nrightarrow 0$. Note that $\int_\Omega\parallel\nabla\times G_n\parallel^2=1/2$ is constant. This implies that we can prevent the pathological example above by adding the curl-free loss term.
> Therefore, the curl-free term is necessary to accurately learn $G$.
>
> If we impose the curl-free loss directly on $G$ without using $\tilde{G}$, we have to take curl on $G$, which is constructed by computing curl on $\Psi$ in (7). However, applying automatic differentiation (AD) consecutively leads to excessive memory consumption and computational inefficiency. In addition, the objective with a high-order derivative using AD has a challenging loss landscape that is difficult to optimize (Wang, 2021).
> We introduced the additional auxiliary variable $\tilde{G}$ to avoid these problems. We conducted an additional experiment by imposing the curl-free loss term directly on $G$ without using $\tilde{G}$. The results are reported in the Table A1 of the attachment. The results shows of the necessity of introducing $\tilde{G}$.
>
> The requirement of $\lambda_1 \rightarrow \infty$ is theoretically important to penalize the constraint in the penalty method. We may gradually increase $\lambda_1$ during training. However, as $\lambda_1$ becomes larger, the balance between the loss terms becomes imbalanced and other terms could be ignored. It means that the condition $u=0$ on $\Gamma$ is not properly enforced and it is also difficult to obtain the curl-free constraint.
>
> **Q4. The qualitative diagrams..**
>
> **Reply**: We summarized quantitative metrics (Chamfer and Hausdorff distances) in the Tables A1, A2, and A3 in the attachment.
> To further investigate the effect of the curl-free term on the learning of $G$, we measure the difference between the given surface normal $n$ and the learned $G$. Given point cloud with normals {$x_i,n_i$},
> we estimate the cosine similarity (CS) $G^Tn:=\frac{1}{N}\sum_{i=1}^N |G(x_i)^T n_i |$, and report it in the Table A4 in the attachment. The results show the angle difference of $G$ and $n$ differing by an average of almost 1.50 (Anchor) and 3.57 (Gargoyle) degrees. Also, $\nabla u$ also has similar differences with $n$. It numerically validates that the curl-free term brings more accurate results to learn $G$ and $u$ for the given test cases. We would like to emphasize that CS does not reflect errors that occur when the 0-level set of $u$ is far from the given point cloud, as it is computed only at points where the $n$ is defined. So, it does not fully describe the quality of the trained surface and gradient fields, but we evaluate it to show the effect of curl-free term on learning accurate gradient fields where the $n$ is defined.
>
> **Q5. What is $F$?..**
>
> **Reply**: Yes, $F=1/3x$ was used in all experiments. In the revised manuscript, we'll make it clear.
>
> **Q6. I am confused why the curl-free constraint needed..**
>
> **Reply**: The necessity of the curl-free constraint is explained in the answer of Q3. In other aspects, for a given $G$ in (10), we agree with the reviewer that $u$ satisfying $G=\nabla u$ is not unique. However, the first term in (8) brings the uniqueness of $u$ such that $G=\nabla u$.
>
> **Q8. The good results of IGR on Thingi10K..**
>
> **Reply**: The similar tendency between IGR and the proposed model seems to be due to the initialization and activation function of the network. Both IGR and our model use the softplus activation and initialize the network to be approximately the SDF of a sphere. On the other hand, SIREN and DiGS use a sine activation and initialize the network in different ways. These are almost the only difference between IGR and SIREN, but the results show that IGR restores smoother surfaces. Therefore, IGR and our model seem to have a tendency to restore smooth surfaces due to these differences.
>
> **Q9. The performance of your model on Daratech..**
>
> **Reply**: A given point cloud of Daratech has an empty part at the back. The proposed model restores the surface by filling in this part, which is why the metric could be high. After the submission of the paper, we have found that $\beta=0.1$ restores this part better. We have reported the additional metric values in Table A2 in the attachment.
>
> **Q10. No clear potential negative societal impact..**
>
> **Reply**: We will specify societal impacts in the revised manuscript as follows: "The proposed PINC allows high-quality representation of 3D shapes only from raw unoriented 3D point cloud. It has many potential downstream applications, including product design, medical imaging, and the film industry. We are aware that accurate 3D surface reconstruction can be used in malicious environments such as  unauthorized reproduction of machines without consent and digital impersonation. However, it is not a work to develop a technique to go to abuse, and we hope and encourage users of the proposed model to concenter on the positive impact of this work."
>
> **Reference**
>
> S. Wang et al. Understanding and mitigating gradient flow pathologies in physics-informed neural networks. SIAM, 2021.

---

> > ### Comment · Reviewer_rqMu · 2023-08-18
> > **Thanks for addressing my concerns, final comments and score change**
> >
> > The new explanation for the theoretical motivation for p-poisson makes a lot more sense, please have that more clear in the paper, it is the most important part of your paper. It would be cool if the uniqueness was able to be shown for your method with a toy problem, e.g. for a very simple set of points in 2D you show that under different initialisations an eikonal loss guided network will converge to drastically different solutions, however with the same initialisations your loss leads to a single (or at much less varied) solution.
> >
> > Same thing with the reason for the curl-free constraint: please improve the explanation in the paper, and it would be great if you can show a toy example showing that often L_2's minimisation does't converge to a curl-free field in practice (while theoretical counter-examples are great, since everything is happening with neural networks which are biased to very smooth approximations due to gradient descent, it would be great to show it is a practical consideration as well). Though its not completely neccessary as your new Table A1 indicates this too.
> >
> > Thanks for providing Tables A1-3, they provide a lot of context about your method. I reccommend having A1 in the main paper alongside the visualisation (maybe A3 as well, and A2 can go to supplementary).
> >
> > As the authors have sufficiently addressed my concerns, I am increasing my score from 4 to 6. I hope they consider my comments for improving the paper (whichever are reasonable for them to do).

---

### Author Rebuttal · Authors · 2023-08-09

**General Response to All Reviewers**

We sincerely thank all the reviewers for their valuable comments, recommendations, and suggestions. The opinions of the reviewers are carefully considered and answering their questions has improved the paper. We first address a common question raised by the reviewers rqMu and ymPA. Then, we address the individual response to each reviewer below. We also attach a supplementary file for Tables. Hopefully, the replies could address all the questions.

$\ $

**Common Question:** Theoretical motivation/intuition for choosing  $p$-poisson instead of eikonal equation to describe SDFs.

**Reply**: The main advantage of using the $p$-Poisson equation is that $u_p$ (solution of (1)) is unique in $W^{1,p}$ (Lindqvist, 2017), which prevents from non-unique weak solution in the eikonal equation $\left\Vert\nabla u\right\Vert=1$. A numerical challenge is to deal with $p \rightarrow \infty$ in order to get a good approximation of the viscosity solution. When the variational formulation (2) is used, the difficulty of using a large $p$ is still persistent numerically. However, it was resolved by using (7), which is one of the main advantages of the proposed algorithm.

**Reference**

P. Lindqvist. Notes on the $p$-Laplace equation. No. 161. University of Jyväskylä, 2017.

$\ $

**Attachment $\downarrow$**

---

### Decision · Program_Chairs · 2023-09-21

**Decision:**

Accept (poster)

**Comment:**

The paper received mixed ratings initially. After rebuttal and reviewer-author discussions, all reviewers became positive. The paper presents a novel PDE-based formulation that can incorporate a variety of constraints. The approach does not require oriented normals as inputs, making it suitable for many application settings. The optimization procedure, which is motivated from variable splitting, is interesting and effective. Therefore, it is recommended to accept the paper due to the technical contributions. Please follow the rebuttal and reviewer's comments to prepare the final version.